# Dispensable players: N-WASP and WASP are not crucial for homology-directed DNA repair

Thu Han Le Phan [ID][1], Anders Buchard[2] & Cord Brakebusch [ID][1][✉]

## Abstract

N-WASP and WASP can induce actin polymerization via Arp2/3 and were reported to be crucial for homology-directed repair (HDR) of DNA double-strand breaks (DSB). The underlying mechanism was suggested to involve nuclear actin polymerization, but the mechanistic details were debated. Unexpectedly, we show now that neither WASP nor N-WASP is required for HDR during CRISPR-mediated genome editing. Using knock-out and over-expression of N-WASP and WASP in U2OS cells, we did not detect alterations in total gene editing, HDR, or the ratio of HDR to non-homologous end joining (NHEJ) as assessed by different methods. Furthermore, we could not observe colocalization of HA-tagged WASP or N-WASP with DSBs. Finally, while the Arp2/3 inhibitor CK-666 and ARPC4 knockdown by siRNA reduced HDR efficiency in U2OS cells, this corresponded with a decreased transfection efficiency and a reduction of the HDR-proficient cell cycle phases S and G2/M. In summary, contrary to expectations, these data do not support a crucial role for N-WASP and WASP in DSB repair.

Subject Categories Cell Adhesion, Polarity & Cytoskeleton; DNA Replication, Recombination & Repair

## Introduction

DNA double-strand breaks (DSBs) are repaired by two major pathways, Non-Homologous End Joining (NHEJ) and Homology-Directed Repair (HDR) (Scully et al, 2019). NHEJ is the dominant pathway, which often results in mutations due to small insertions or deletions at the damage sites (Pannunzio et al, 2018; Weterings and Van Gent, 2004). HDR, on the other hand, involves complementary base pairing and can be divided into different subtypes (Hartlerode and Scully, 2009; Sung and Klein, 2006). HDR is initiated by end-resection, which creates single-stranded DNA on both sides of the DSB. If this single-stranded region is short, base pairing of single-stranded DNA of the DSB ends results in repair by Microhomology Mediated End Joining (MMEJ) and short deletions (Sfeir et al, 2024;

Truong et al, 2013). If end-resection is more extensive, the ssDNA is bound and stabilized by RPA. Replacement of RPA by RAD51 enables the repair by Homologous Recombination (HR) involving a double-stranded DNA template with homologous sequences (Liu et al, 2010; Sugiyama et al, 1997). Alternatively, in the presence of a single-stranded homologous DNA template, replacement of RPA by RAD52 will allow repair by Single-Strand Templated Repair (SSTR) (Kan et al, 2017; Paulsen et al, 2017; Shao et al, 2017). This pathway is frequently used for CRISPR/Cas-mediated precise gene editing due to its high efficiency, and optimizing its efficiency is of high importance for the development of CRISPR/Cas-based gene therapies in the clinic (Gallagher and Haber, 2021; Richardson et al, 2018).

Neural Wiskott–Aldrich syndrome protein (N-WASP) and Wiskott–Aldrich syndrome protein (WASP) are highly homologous actin nucleation-promoting factors that are present both in the cytoplasm and nucleus (Kramer et al, 2022; Sarkar et al, 2014; Suetsugu and Takenawa, 2003; Taylor et al, 2010). In their open conformation, they mediate branched actin polymerization via binding to G-actin and Arp2/3 complex (Higgs and Pollard, 2000; Kim et al, 2000). Due to their role in actin polymerization, N-WASP and WASP are involved in endocytosis and cell migration (Badour et al, 2007; McGavin et al, 2001; Merrifield et al, 2004). In addition, both molecules act as epigenetic regulators in the nucleus (Chandnani et al, 2023; Li et al, 2019; Li et al, 2018; Taylor et al, 2010; Yuan et al, 2022). Several recent studies indicated that WASP contributes to DNA repair through HDR (Han et al, 2022; Schrank et al, 2018; Zagelbaum et al, 2023). Although different mechanisms have been proposed, most of those involve the promotion of nuclear actin polymerization. Notably, WASP is only present in non-erythroid hematopoietic tissues, while N-WASP is expressed ubiquitously (Takenawa and Suetsugu, 2007; Zhu et al, 1997). This led us to speculate that N-WASP may facilitate HDR in cells lacking WASP. An earlier study furthermore suggested that all molecules that promote nuclear actin polymerization, including N-WASP, can promote the binding of RPA to single-stranded DNA, an important step in HDR (Iftode et al, 1999; Nieminuszczy et al, 2023; Zou and Elledge, 2003). In addition, both Arp2/3-dependent and Arp2/3-independent nuclear actin polymerization have been found to be crucial for HDR (Caridi et al, 2019; He and Brakebusch, 2024). In a previous study, we demonstrated in a transient overexpression system that N-WASP promotes the formation of nuclear nodules of F-actin in a cortactin-dependent manner, confirming the potential of N-WASP to encourage nuclear actin polymerization (Jiang et al, 2025).

In this study, we examined the roles of WASP and N-WASP in the DNA damage response using U2OS cells. Surprisingly, we

[1]Biotech Research and Innovation Center (BRIC), University of Copenhagen, Copenhagen, Denmark. [2]Section of Forensic Genetics, Department of Forensic Medicine, University of Copenhagen, Copenhagen, Denmark. [✉]E-mail: cord.brakebusch@bric.ku.dk

found no evidence to support a role for N-WASP or WASP in HDR using knock-out (KO) and overexpressing systems. Despite numerous publications highlighting the relationship between nuclear actin polymerization factors and DNA repair, the significance of this connection remains unclear.

# Results

## Loss of N-WASP does not affect the proliferation, cell cycle, or migration of U2OS cells

To investigate the role of N-WASP in the DNA damage response (DDR), we generated different polyclonal N-WASP knock-out cell lines (U2OS, HEK293T, and MCF-10A) using a lentiviral vector encoding Cas9 and a single guide RNA (gRNA) targeting exon 1 or 2 of the N-WASP gene. Cells transduced with a lentiviral vector expressing a non-targeting gRNA (ntgRNA) were established as a control. Western blot analysis revealed efficient deletion of N-WASP (Fig. 1A).

As an indirect marker for functionally relevant changes in cytoplasmic actin polymerization, we investigated cell migration in a wound-healing assay. N-WASP was shown earlier to play a key role in modulating migration in various cancer cell lines (Chung et al, 2022; Kim et al, 2024; Takahashi and Suzuki, 2011; Yu et al, 2012). Consistent with previous reports, N-WASP knock-out U2OS cells showed significantly slower wound-closure rates compared with their wild-type (ntgRNA) controls (Fig. 1B), confirming both the functional role of N-WASP and the successful generation of the knock-out cell line.

Given that proliferating cells are more susceptible to DNA damage and the DNA damage response is intricately linked to cell cycle regulation (Hustedt and Durocher, 2017; Kaufmann and Paules, 1996; Ou and Schumacher, 2018), we investigated whether knock-out of N-WASP affects cell cycle progression of U2OS, HEK293T, and MCF-10A cells. Cell cycle profiles were assessed by EdU incorporation combined with DAPI staining, enabling a precise discrimination of the cell cycle stages. Yet, quantitative analysis did not reveal a significant difference in cell cycle progression of ntgRNA control and N-WASP knock-out cells in all three cell lines (Fig. 1C). This indicates that the loss of N-WASP does not indirectly affect DNA repair efficiency through alterations in cell cycle regulation.

## Loss of N-WASP does not affect the repair of ionizing radiation-induced double-strand breaks

Histone H2AX phosphorylation at serine 139 (γH2AX) is a sensitive biomarker for DSB (Rogakou et al, 1998) and DNA repair leads to its attenuation (Chowdhury et al, 2005). Therefore, alterations in the foci numbers and the intensities of γH2AX can serve as indicators of an aberrant DNA damage response. Kinetic analysis of U2OS, HEK293T, and MCF-10A cells exposed to a 2 Gy dose of ionizing radiation demonstrated that the loss of N-WASP did not lead to increased single-nucleus γH2AX foci numbers, neither before irradiation nor at 1, 2, 6, or 24 h after irradiation in any cell cycle phase (Figs. 2A and EV1). Interestingly, we observed in N-WASP KO HEK293T cells significant reductions in γH2AX foci numbers at 1 and 2 h post irradiation at all cell cycle phases (Fig. 2A).

Regarding the γH2AX intensities, we observed no significant differences in nuclear γH2AX mean intensities between the wild-type and N-WASP knock-out U2OS, HEK293T, and MCF-10A at all time points post-irradiation in either S or G2/M phase (Fig. 2B). In sham-irradiated controls (0 h), however, we detect a slight but significant increase in nuclear γH2AX mean intensity in U2OS N-WASP knock-out cells compared with their wild-type (ntgRNA) counterparts (Fig. 1B). Because elevated baseline γH2AX could indicate genomic instability (Georgoulis et al, 2017; Yu et al, 2006), we further examine the percentage of cell carrying micronuclei in all given cell lines before and 24 h after exposure to 2 Gy X-ray irradiation. The results showed no significant differences in the proportion of micronuclei-positive cells between wild-type and N-WASP knock-out cell lines (Fig. 3), suggesting that the increased baseline γH2AX intensity in G1-phase U2OS cells is attributable to factors other than altered genome instability.

In summary, these findings do not reveal a crucial role of N-WASP in DNA repair.

## Neither N-WASP nor WASP is colocalizing with γH2AX at DSBs

WASP, which is expressed only in hematopoietic cells, and N-WASP, which is ubiquitously expressed, share significant structural and functional similarities (Kramer et al, 2022). Since WASP was previously reported to assemble into foci at DSB sites (Schrank et al, 2018), we investigated whether this is also the case for N-WASP. To do that, we generated a U2OS cell line lacking endogenous N-WASP and overexpressing HA-N-WASP (Fig. 4A) and exposed these cells to 100 nM camptothecin (CPT) in 1 h. CPT is a topoisomerase I inhibitor that is able to induce DNA double-strand breaks primarily repaired by HDR (Ferrara and Kmiec, 2004; O'Connell et al, 2010; Robison et al, 2005). We co-stained HA-N-WASP, detected by antibodies against the HA tag, and the DSB marker γH2AX. However, a near-zero Pearson's correlation coefficient of nuclear HA-N-WASP and DSB indicated no correlation between the two signals, suggesting no significant localization of N-WASP at DSB foci (Fig. 4C). For comparison, in the positive control cells that were co-stained for γH2AX and the HDR marker BRCA1, distinct BRCA1 foci were readily visible. These foci colocalized strongly with γH2AX, as most cells exhibited moderate to very high correlation values ($R \geq 0.5$), with more than 60% falling within the high-correlation range ($R = 0.6$–$0.8$) (Fig. 4C). Surprisingly, we neither did observe a colocalization of HA with γH2AX in U2OS cells overexpressing HA-WASP (Fig. 4B,C), contrary to the expectations. In addition, we observed almost no discernible nuclear foci of HA–WASP at DSB sites, in contrast to previous reports, and most of the detected HA–WASP signal was localized to the cytoplasmic or peri-nuclear regions.

To reduce staining of molecules not strongly bound to chromatin, we performed a pre-extraction with Triton X-100 before staining for γH2AX and HA-N-WASP. While the pre-extraction only slightly reduced the staining for γH2AX, it strongly decreased nuclear staining for HA-N-WASP, indicating that it is not strongly associated with the chromatin (Fig. 4D).

This prompted us to examine the specificity of the WASP antibody used to detect the colocalization of WASP with γH2AX in the previous study. We therefore generated WASP knock-out U2OS cells by CRISPR/Cas9 and U2OS cells transduced with HA-

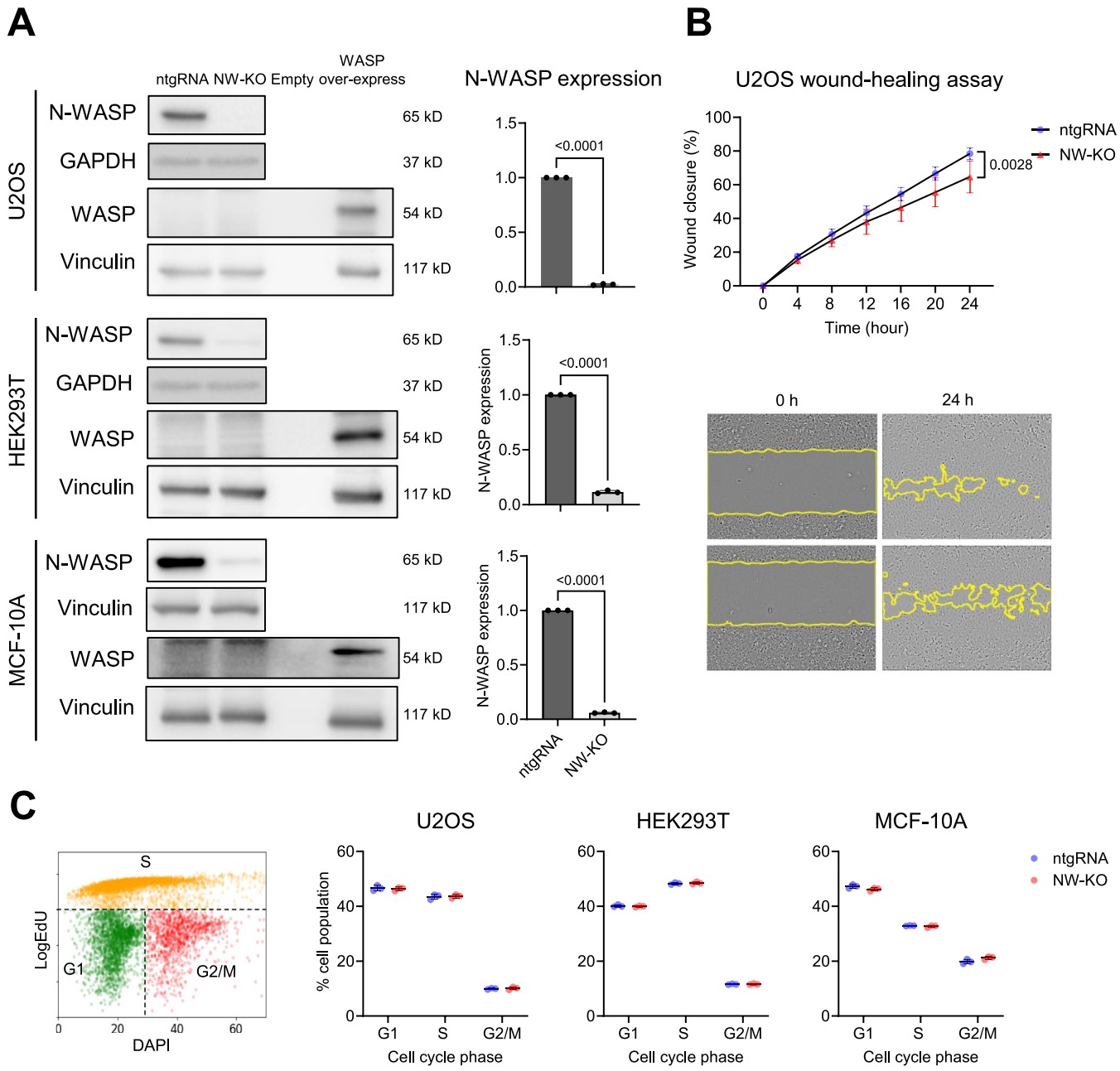

**Figure 1. N-WASP knock-out cell lines show no compensatory expression of WASP and no alteration in cell cycle.**

(A) Immunoblot analysis of cell lines transduced with lentiviral vectors expressing Cas9 and either a non-targeting gRNA (ntgRNA; control) or an N-WASP–specific gRNA (NW-KO). Quantification of N-WASP protein levels is shown on the right. Potential compensatory upregulation of WASP was assessed, and correct WASP identification was verified using a HA-WASP-overexpressing positive control. (B) Wound-closure kinetics of control (ntgRNA) and N-WASP knockout U2OS cells, expressed as the percentage of wound area closed at 0, 4, 8, 12, 16, 20, and 24 h post-scratching. Each data point represents one biological experiment comprising 18–46 technical replicates. (C) Cell-cycle profiling based on EdU incorporation and DAPI intensity. Cells were incubated with 10 μM EdU for 30 min prior to immunofluorescence staining. Left: gating strategy used to distinguish G1, S, and G2/M populations. Right: distribution of cell-cycle phases in control and N-WASP knockout cells. Data are presented as mean ± SD, and statistical significance was assessed using a two-tailed unpaired $t$ test. Source data are available online for this figure.

WASP. At the genomic level, the knock-out efficiency of WASP was investigated by Sanger sequencing of the genomic PCR around the gRNA cut site. Both DECODR and ICE analyses confirmed a successful knock-out, with efficiencies of 95.2% (frameshift mutation, DECODR) and 92% (knock-out score, ICE) (Fig. EV2A).

At the protein level, western blot revealed multiple bands detected by the WASP antibody used earlier. Expression of HA-tagged WASP resulted in a strong band at the expected molecular weight of about 60 kD, indicating that the antibody indeed recognizes WASP. However, we could not detect a clear band at the expected

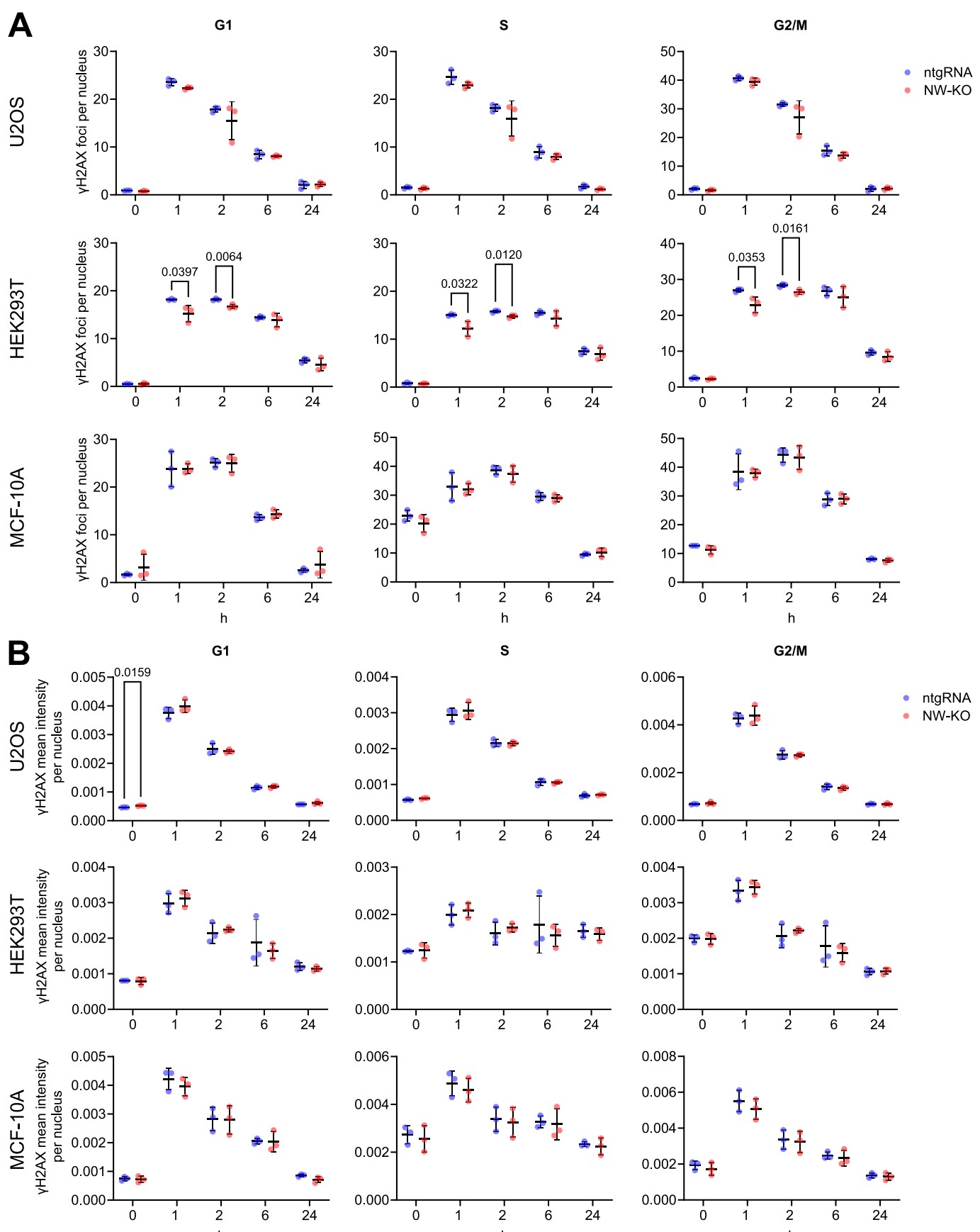

**Figure 2.  Loss of N-WASP is not increasing gH2AX foci numbers and intensities after irradiation.**

Cell-cycle-dependent γH2AX (**A**) foci numbers and (**B**) mean intensities per nucleus measured at the indicated time points following exposure to 2 Gy X-ray irradiation; Sham-irradiated cells served as the 0 h controls. Cells were incubated with 10 μM EdU for 30 min prior to collection. Cell-cycle phases were assigned based on nuclear EdU and DAPI intensities. Each data point represents one experiment, with more than 6000 nuclei analyzed per sample. Data are presented as mean ± SD, and statistical significance was determined using a two-tailed unpaired $t$ test.

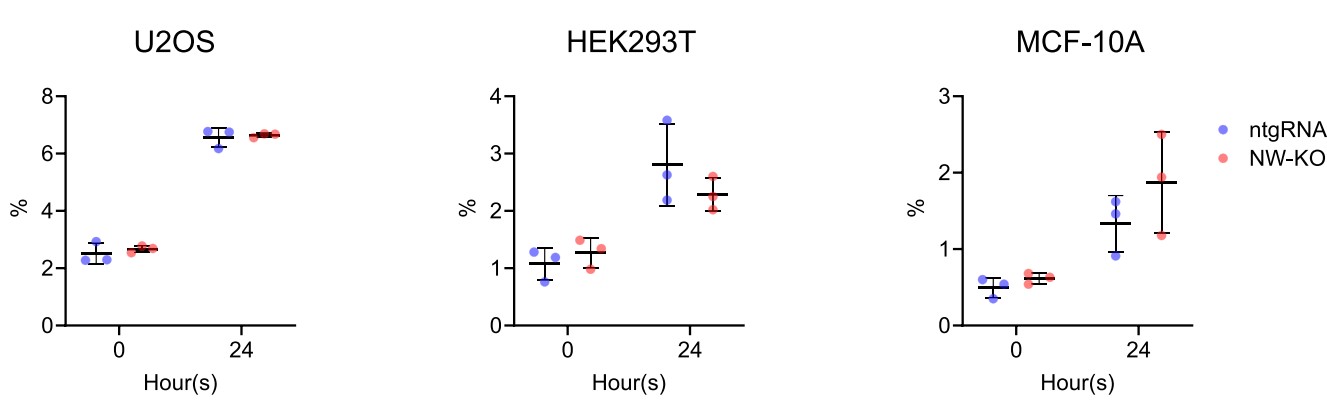

% cell population with micronuclei

**Figure 3.  Loss of N-WASP is not increasing frequency of micronuclei.**

The proportions of cell population with at least 1 micro nucleus in wild-type and N-WASP knock-out U2OS, HEK293T and MCF-10A cell lines were measured before and 24 h after being subjected to 2 Gy X-ray irradiation. Each data point represents one experiment, with more than 6000 nuclei analyzed per sample. Data are presented as mean ± SD, and statistical significance was determined using a two-tailed unpaired $t$ test.

size of WASP in U2OS ntgRNA control cells. Moreover, we could not detect any protein band that was lost in the WASP knock-out compared to the controls (Fig. EV2B). Instead, many non-specific bands were observed, some with even higher intensities than the HA-WASP band, suggesting a significant cross-reactivity of the WASP antibody with other proteins. Thus, in contrast to earlier reports, we could not confirm the expression of WASP in the non-hematopoietic U2OS cells.

Next, we performed immunofluorescent staining with antibodies against WASP and HA-tag in U2OS cells that overexpress HA-tagged WASP, with WASP knockout cells serving as the negative control. In WASP knock-out cells, which did not show detectable WASP level in Western Blot, we observed strong nuclear staining with the WASP antibody, suggesting false positive signals. The expression of HA-WASP resulted in an additional, mainly cytoplasmic signal colocalizing with HA (Fig. EV2C). Quantification showed that HA-WASP–expressing cells exhibited a strong increase in HA signal intensity, allowing clear discrimination from background staining observed in the knockout control (Fig. EV2D). Although the WASP antibody was able to detect genuine WASP, as suggested by its overlapping with HA signals, overexpression of HA-WASP did not result in a significant increase in the WASP antibody signal, indicating a high amount of non-specific nuclear staining (Fig. EV2D). Furthermore, the WASP antibody signal had a much higher nuclear fraction than the HA antibody, although a similar distribution should be detected in the case of a specific WASP antibody (Fig. EV2E). These data indicate that results obtained with the tested antibody against WASP should be

interpreted with caution and controlled for specificity by CRISPR-mediated WASP KO.

## Neither absence nor overexpression of WASP changes HDR efficiency in U2OS cells

Although neither N-WASP nor WASP presents at the DSB sites, they could still influence the HDR process. To investigate whether these two proteins could affect HDR efficiency, we employed two different CRISPR-based HDR assays: the Indel Detection by Amplicon Analysis (IDAA) and the mClover-LMNA reporter assay. In both assays, DSBs were generated by transfecting cells with a plasmid encoding Cas9 and an endogenous gene-targeting gRNA.

In the mClover-LMNA assay, an additional donor plasmid served as the repair template for in-frame insertion of an mClover tag into the LMNA locus. Cells that underwent HDR therefore exhibit fluorescent nuclei and could be detected by flow cytometry (Pinder et al, 2015). A plasmid encoding iRFP670 was co-transfected for the transfection control, and the HDR efficiency was calculated as the proportion of mClover-positive cells among the iRFP670-positive population at day 3 post-transfection.

For the IDAA assay, the Cas9/gRNA expressing vector contained an EGFP marker, allowing selection of transfected cells. A single-stranded DNA donor containing an 8-bp asymmetric insertion flanked by two 50-bp homology arms was used as the HDR template. Two days after transfection, EGFP-positive cells were sorted, and the length of PCR amplicons spanning the targeted genomic region was determined by IDAA, identifying

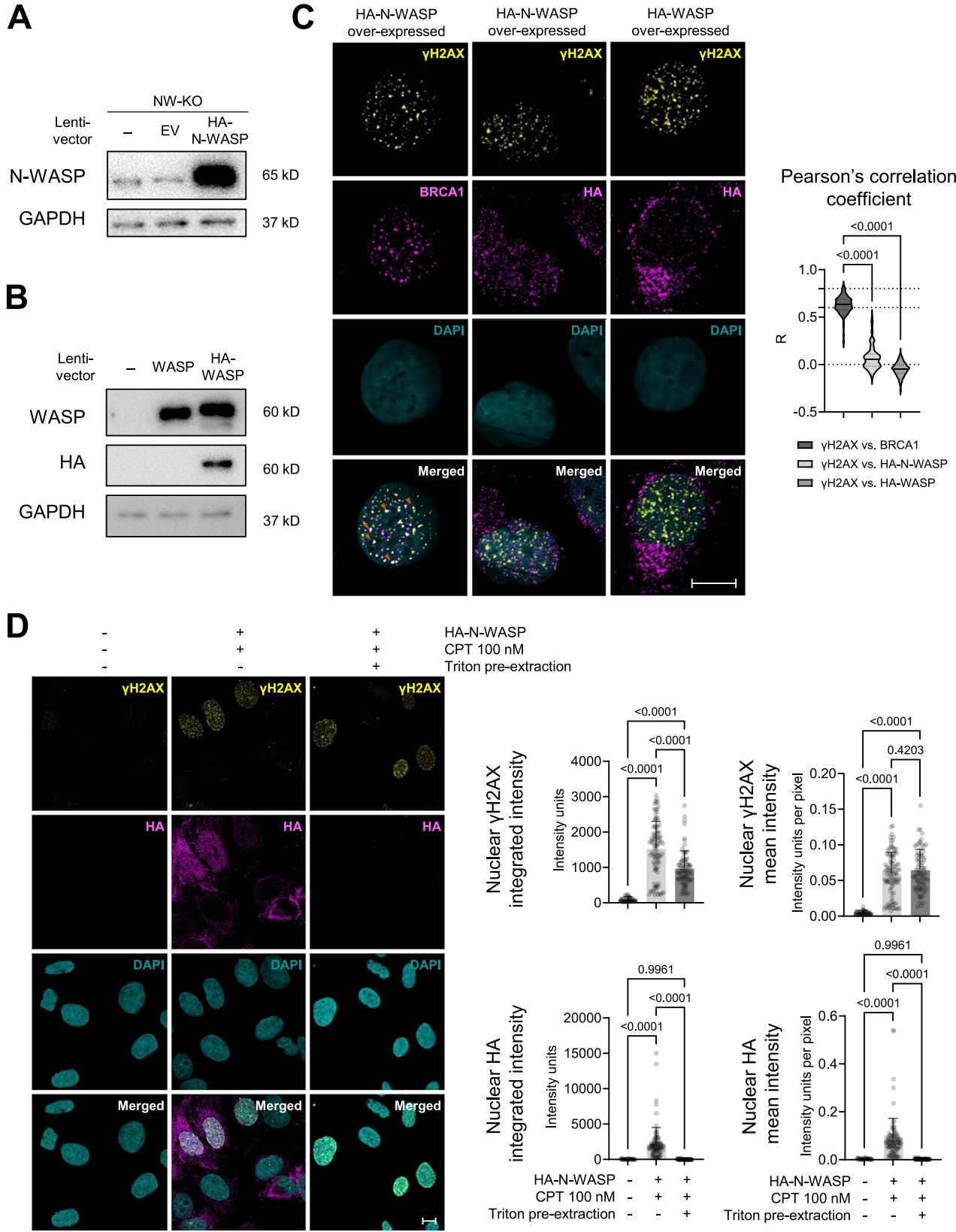

**Figure 4. N-WASP or WASP are not colocalizing with camptothecin-induced gH2AX foci.**

(A) Immunoblotting of N-WASP knock-out U2OS cells (sgRNA2) lentivirally transduced HA-N-WASP or an empty vector (EV). (B) Immunoblotting of wild-type U2OS cells lentivirally transduced with WASP, HA-WASP, or empty vector (EV). (C) U2OS cells transduced with HA-N-WASP, HA–WASP or empty vector (EV), were treated with 100 nM camptothecin (CPT) for 1 h to induce DNA damage and stained by immunofluorescence as indicated. Left: representative images with arrows indicating examples of colocalization between the relevant signals (scale bar: 10 μm). Right: colocalization of γH2AX with BRCA1, HA–N-WASP, or HA–WASP was quantified using Pearson's correlation coefficient (R). Each dot represents a nucleus, with at least 86 nuclei analyzed per condition. Data are presented as violin plots showing the median and interquartile range. (D) U2OS cells with or without HA–N-WASP were treated with either 0.1% DMSO or 100 nM CPT for 1 h to induce DNA damage. Cells were subsequently processed with or without 0.1% Triton X-100 prior to immunofluorescence staining. Left: Representative images (scale bar: 10 μm). Right: Integrated nuclear intensities of γH2AX and HA. Values are presented as mean ± SD, with at least 33 nuclei analyzed per group. Statistical significance was assessed by one-way ANOVA with Tukey's correction for multiple comparisons. Source data are available online for this figure.

unmodified (wt length), HDR (+8 bp), and NHEJ (all other changes of amplicon length) (Bischoff et al, 2023; Yang et al, 2015).

WASP KO did not affect transfection efficiency in U2OS cells (Fig. 5A). Importantly, we did not observe any significant differences in HDR-mediated LMNA repair in the mClover assay, nor in total gene editing, HDR, NHEJ, or the HDR:NHEJ ratio for PAX2 or TSPAN12 in the IDAA assay when comparing WASP knockout cells with control wild-type cells (Fig. 5B,C). These findings indicate that either U2OS cells do not express WASP, hence confirming our earlier results, or that WASP is not required for HDR.

For further examination, we performed both assays on U2OS cells overexpressing WASP. We tested both WASP and HA-WASP (Fig. 4B), to investigate a potentially decreased functionality WASP in the presence of HA-tag at its N-terminus. Also, overexpression of WASP had no influence on the transfection efficiency (Fig. 6A). Consistent with the knockout results, neither WASP nor HA-WASP overexpression altered HDR efficiency in the mClover-LMNA assay or affected total editing, HDR, or the HDR:NHEJ ratio at the PAX2 or TSPAN12 loci in the IDAA experiments (Fig. 6B,C).

In summary, our results indicate no important role of WASP in HDR.

## Absence of N-WASP does not significantly change HDR efficiency

Next, we compared the HDR efficiency of N-WASP knock-out (NW-KO) and ntgRNA control cells using the mClover-LMNA assay and IDAA in U2OS and HEK293T cells. Loss of N-WASP protein did not increase expression of WASP in either of the two tested cell lines (Fig. 1A), ruling out any compensatory effect of WASP. However, loss of N-WASP slightly increased transfection efficiency (Fig. EV3A,B).

In U2OS, N-WASP knock-out cells showed comparable HDR efficiency to the wild-type control for both the TSPAN12 locus in the IDAA assay and the LMNA locus in the mClover assay (Fig. 7A,B). Although we observed a significant increase in HDR efficiency in the PAX2 locus, this effect does not support the role of N-WASP in promoting HDR.

In HEK293T cells, IDAA analysis of both target genes revealed no significant differences in total editing, HDR, or the HDR:NHEJ ratio between N-WASP knockout cells and their wild-type counterparts (Fig. 7A). However, in the mClover-LMNA assay, N-WASP knockout cells displayed a slight but statistically significant reduction in HDR efficiency. Notably, whereas U2OS cells exhibited similar HDR efficiencies across all three tested loci, the HDR efficiency for LMNA in the mClover assay was only around 1%, which is markedly lower than the approximately 25% HDR efficiencies observed for PAX2 and TSPAN12 in the IDAA assay.

Deficiency in HR related genes often results in synthetic lethality by increased sensitivity to PARP inhibitors (Murai and Pommier, 2023). However, N-WASP KO U2OS or HEK293T cells showed no decreased survival in the presence of PARPi compared to control cells (Fig. 7C).

These data do not support an important role for N-WASP in promoting HDR-

## Arp2/3 inhibition reduces transfection efficiency, S-phase, and HDR

Arp2/3-mediated actin polymerization was reported to play a role in HDR. Since N-WASP can promote Arp2/3-dependent nuclear actin polymerization, as we recently showed, we wanted to test the role of Arp2/3-dependent actin polymerization on DNA repair in the U2OS cells investigated by us for N-WASP function.

First, we tested whether Arp2/3 inhibition affects the cell cycle of U2OS, MCF-10A, and HEK293T cells. Incubation of these cell lines for 24 h with 100 μM of the Arp2/3 inhibitor CK-666 significantly increased G1 and decreased S and G2/M, particularly in HEK293T cells (Fig. 8A). Inhibition of Arp2/3 function by siRNA against ARPC4 in U2OS cells also showed a significant decrease in G1 and an increase in S phase cells, although to a lesser extent than observed with CK-666 (Fig. 8B). CK-666 treatment in addition significantly reduced transfection efficiency in U2OS and HEK293T cells (Fig. 8C). These effects of Arp2/3 inhibition need to be considered when analyzing its effect on DNA repair.

Next, we assessed how CK-666 is affecting the cell cycle after irradiation with 2 Gy. Also, in this setting, we observed some significant cell cycle differences between treated and untreated cells (Fig. 9A). Assessing irradiation-induced DNA damage by cell cycle-resolved γH2AX staining, we detected increased γH2AX foci after 1 h and a slightly increased mean fluorescence per pixel in nuclei in S-phase after 6 h in CK-666 treated cell (Fig. 9B,C).

To further study the effects of CK-666 on DNA repair, we performed IDAA- and mClover -based repair assays. In U2OS, we detected a significant decrease in HDR efficiency of DSB repair of the PAX2 or TSPAN12, while the total gene editing efficiency was unaltered (Fig. 10A). In HEK293T cells, on the other hand, HDR of TSPAN12 was not significantly changed by CK-666 treatment while the total editing efficiency was significantly decreased (Fig. 10A). When targeting the LMNA gene by the mClover-LMNA assay, which uses a double-stranded DNA repair template, CK-666 showed no effect in U2OS, but a significant reduction in HDR efficiency HEK293T cells (Fig. 10B). Inhibition of Arp2/3 by ARPC4 siRNA on U2OS displayed a slight but significant reduction of HDR as measured by the mClover assay (Fig. 10C).

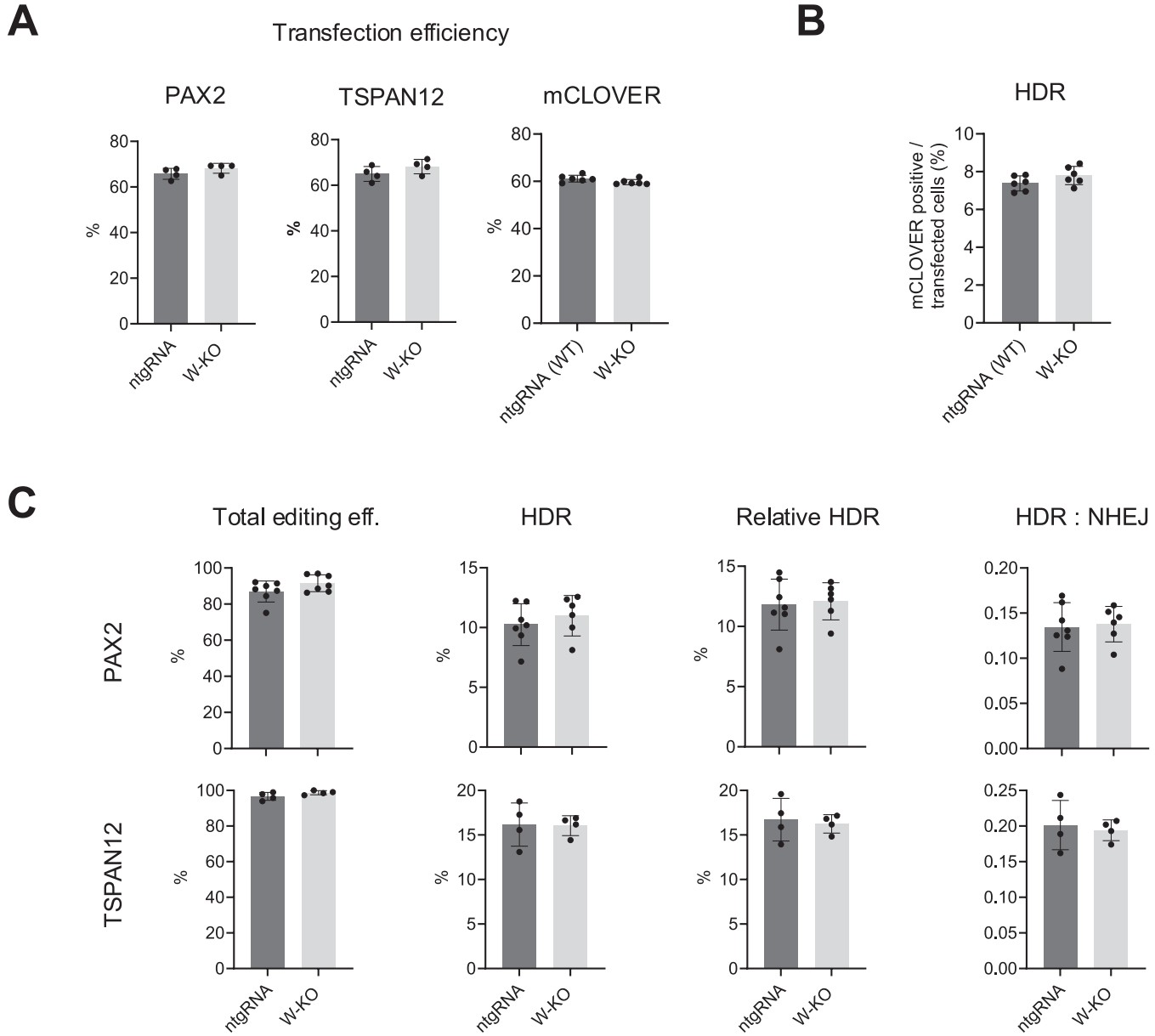

**Figure 5.   WASP KO in U2OS cells does not affect DNA repair.**

WASP KO U2OS cells were transfected with Cas9 and an sgRNA targeting indicated endogenous genes (TSPAN12, PAX2) and corresponding single-stranded repair templates for IDAA experiments, or with Cas9 and an sgRNA for LMNA and a double-stranded repair plasmid inserting the fluorescent mClover. (A) Transfection efficiency as determined by FACS for fluorescent transfection marker. (B) HR mediated insertion of mClover in indicated cell types. Each dot represents one independent experiment with 10,000 transfected cells (for mClover-LMNA assay) or 10,000 live cells (for IDAA assay) recorded. (C) Total editing, HDR efficiency, and HDR:NHEJ ratio in control (ntgRNA), WASP knock-out (W-KO), and N-WASP knock-out (NW-KO) U2OS cells after Cas9-induced DSBs were measured by IDAA. Each dot represents one independent experiment. Values are shown as mean ± SD, with significance assessed by a two-tailed unpaired *t* test. Source data are available online for this figure.

These results suggest a subtle role of Arp2/3-mediated actin polymerization in DNA repair.

## Discussion

The results presented in this study do not support a major functional role for WASP or N-WASP in the DNA damage response, as evidenced by their absence at DNA damage sites as detected by immunofluorescent staining and the mostly unchanged DNA damage marker γH2AX levels and NHEJ and HDR efficiencies in cells with a KO or overexpression of either protein. These data are important because they contradict earlier studies on this subject. Non-specific nuclear binding of a WASP antibody used in previous investigations might explain some of these differences.

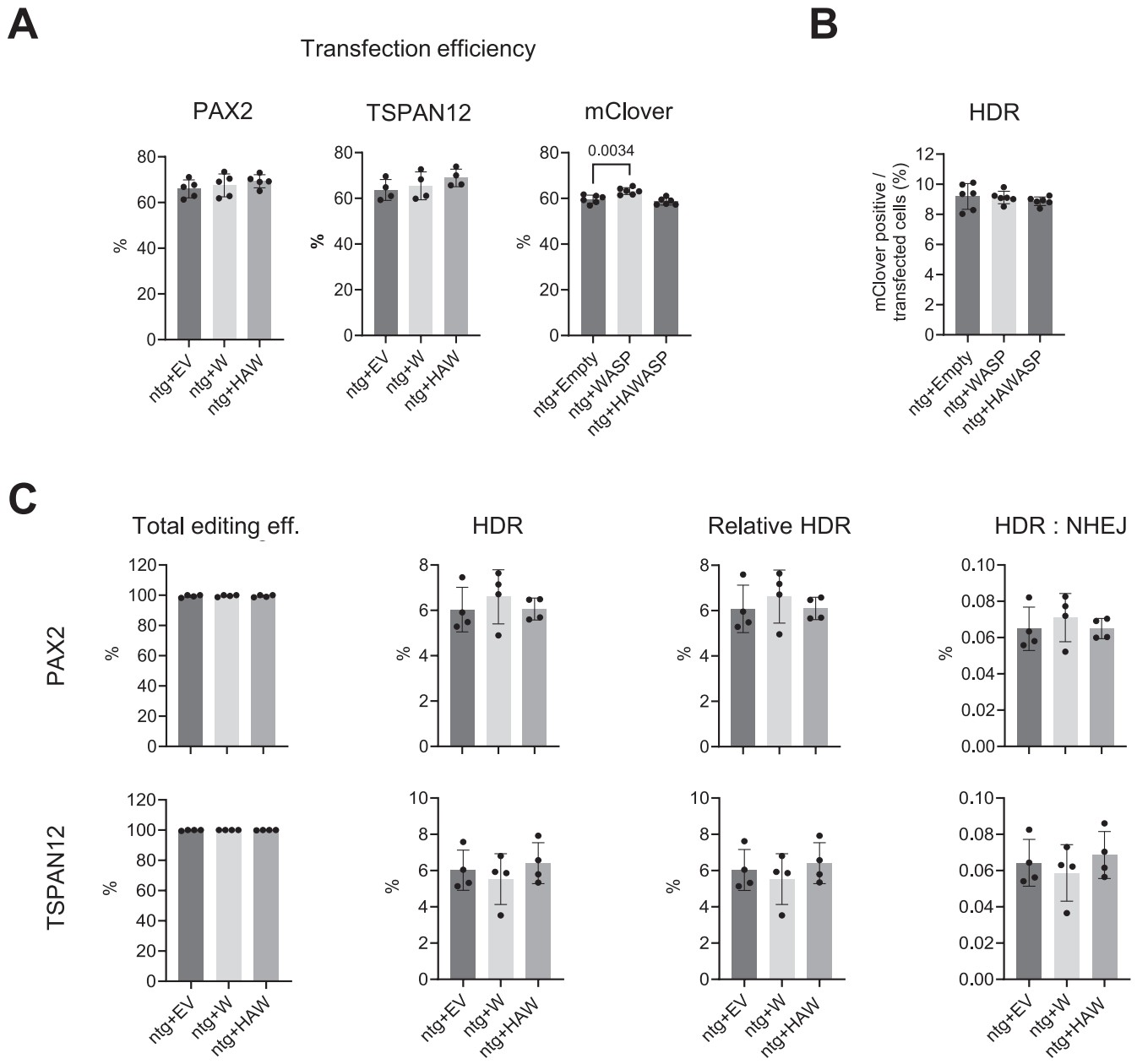

**Figure 6. WASP overexpression in U2OS cells does not affect DNA repair.**

Control U2OS cells (ntg) transduced with either WASP (W), HA-WASP (HAW), or an empty vector were transfected with plasmids encoding Cas9 and an sgRNA targeting indicated endogenous genes (TSPAN12, PAX2) and corresponding single-stranded repair templates for IDAA experiments, or with Cas9, an sgRNA for LMNA, and a double-stranded repair plasmid inserting the fluorescent mClover. (**A**) Transfection efficiency as determined by FACS for fluorescent transfection marker. (**B**) HR-mediated insertion of mClover in indicated cell types. (**C**) Total editing, HDR efficiency, and HDR:NHEJ ratio in indicated cell types as measured by IDAA. Each dot represents one independent experiment. Values are shown as mean ± SD, with significance assessed by one-way ANOVA with Tukey correction for multiple comparisons tests. Source data are available online for this figure.

Schrank et al showed nuclear foci formation of WASP colocalizing with gH2AX in U2OS osteosarcoma cells and mouse tail fibroblasts after treatment with the radiomimetic neocarzinostatin (Schrank et al, 2018). However, neither of these two non-hematopoietic cell types is expected to express any WASP protein, which is restricted to the blood lineage. Accordingly, we could not detect WASP protein neither in our U2OS cells nor in MCF-10A or

HEK293T cells. Furthermore, we could demonstrate that the antibody used for immunofluorescent detection of WASP is not very specific in Western blot and clearly shows a strong non-specific nuclear staining in WASP-deficient U2OS cells, indicating that it is a problematic tool. In addition, we could not observe colocalization of nuclear foci of WASP or N-WASP with γH2AX after exposure to camptothecin using HA-tagged WASP. These

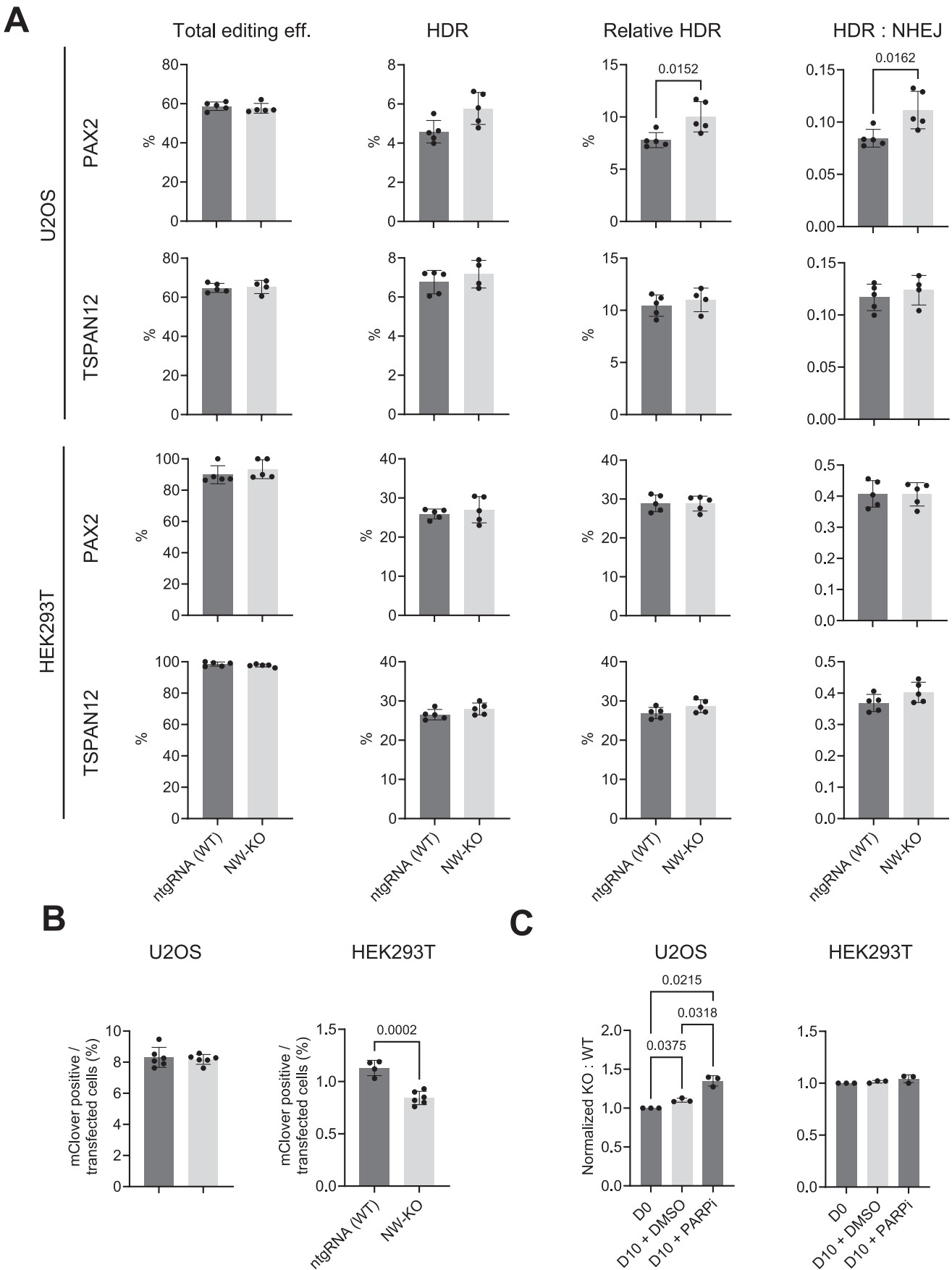

◄ **Figure 7. Loss of N-WASP does not reduce HDR in U2OS cells.**

Indicated cell types with (ntgRNA) and without N-WASP (NW-KO) were transfected with plasmids encoding Cas9 and an sgRNA targeting indicated endogenous genes (TSPAN12, PAX2) and corresponding single-stranded repair templates for IDAA experiments, or with Cas9, an sgRNA for LMNA, and a double-stranded repair plasmid inserting the fluorescent mClover. (A) Total editing, HDR efficiency, and HDR:NHEJ ratio in control (ntgRNA) and N-WASP KO (NW-KO) in the indicated cell lines as measured by IDAA. Each dot represents one independent experiment. (B) HR-mediated insertion of mClover in indicated cell types. Each dot represents one independent experiment. (C) Competitive growth assay of indicated cell lines with or without KO of N-WASP in the presence and absence of PARP inhibitor (Talazoparib, 10 nM). Shown is the ratio of N-WASP KO to WT before (D0) and after 10 days culture (D10). Values are shown as mean ± SD, with significance assessed by two-tailed unpaired *t* test (A, B) or one-way ANOVA with Tukey correction for multiple comparisons tests. Source data are available online for this figure.

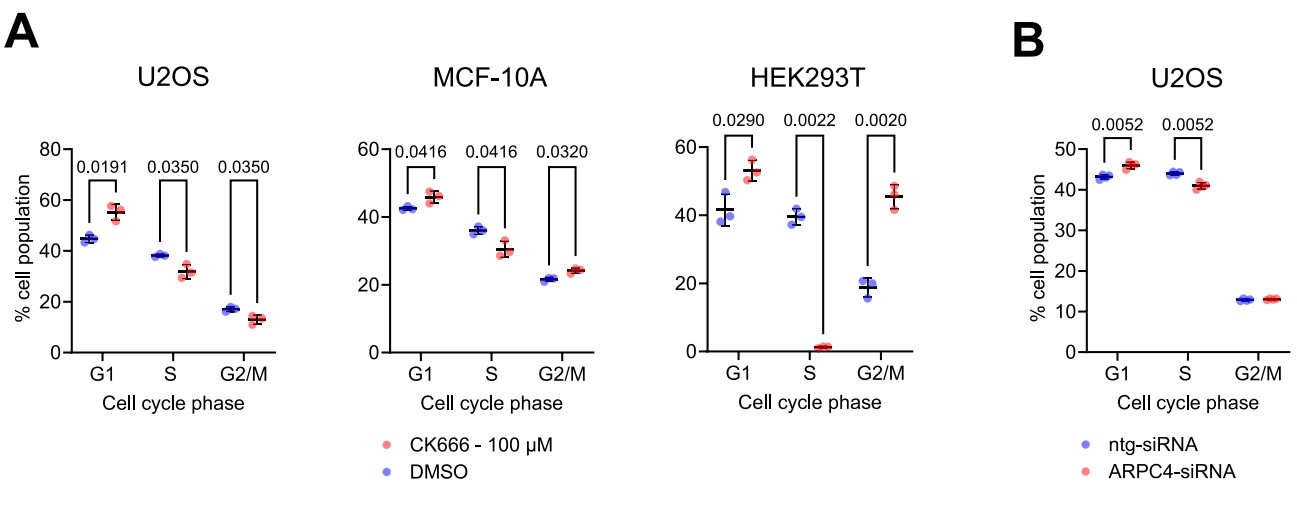

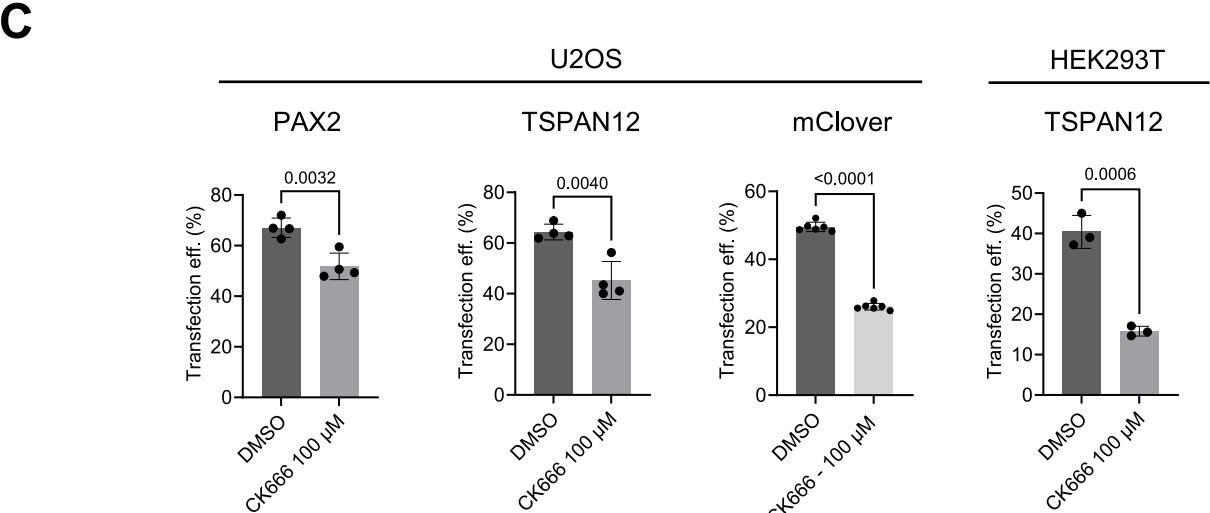

**Figure 8. Arp2/3 inhibition alters cell cycle and reduces transfection efficiency.**

(A) Indicated cell lines were treated for 24 h with DMSO or 100 μM CK666 and then analyzed for cell cycle by EdU incorporation assay. (B) U2OS cells were transfected with either non-targeting or ARPC4-specific siRNA for 48 h, then subjected to cell cycle analysis. (C) Indicated cells with specific plasmid transfection were pre-treated for 1 h with 100 μM CK666 and then checked for transfection efficiency by the corresponding fluorescent marker proteins. Values are shown as mean ± SD, with significance assessed by two-tailed unpaired *t* test. Source data are available online for this figure.

immunofluorescent data, however, cannot exclude a small amount of WASP or N-WASP at the DSB, which might be detectable with a proximity ligation assay.

Schrank et al showed further that treatment of U2OS cells with the Arp2/3 inhibitor CK-666 reduces HDR efficiency in an I-SceI-based GFP reporter system. In this system, the repair template is double-stranded DNA integrated into the genome. Using a single-stranded DNA repair template, we also observed reduced HDR efficiency in CK-666-treated U2OS cells. In contrast to Schrank et al, however, we detected a significantly decreased S and G2/M

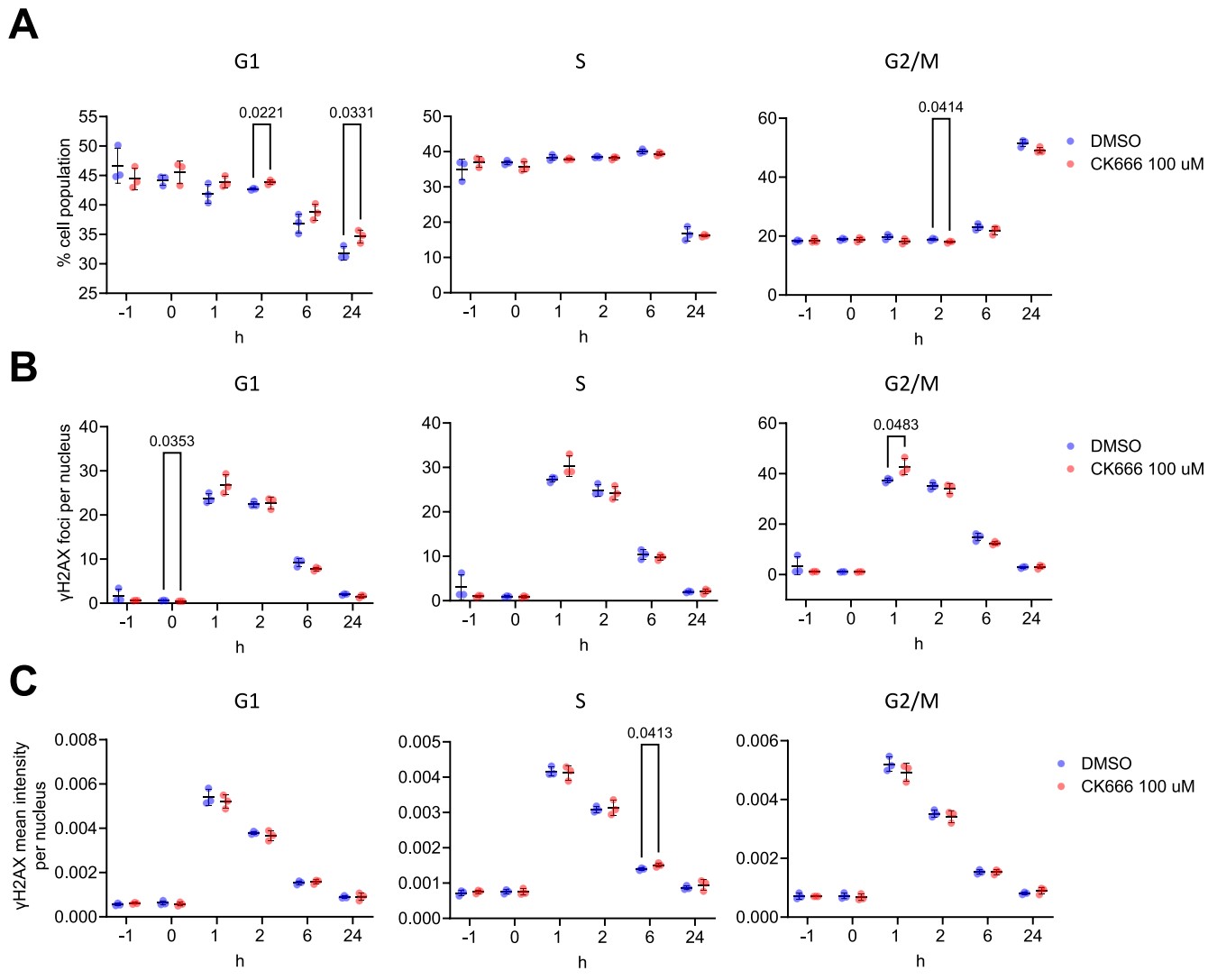

**Figure 9. The Arp2/3 inhibitor CK666 shows subtle effects on irradiation induced γH2AX formation.**

Following 1 h treatment with 100 μM CK666 (−1 h), U2OS cells were irradiated with 2 Gy X-ray and analyzed at the indicated time points for γH2AX by immunofluorescent staining and for cell cycle by EdU incorporation. Sham-irradiated cells were used as controls (0 h). (**A**) Cell cycle analysis and (**B**) cell cycle-dependent γH2AX foci numbers and (**C**) γH2AX mean intensity per nucleus. Each dot represents one experiment with over 6000 nuclei analyzed per sample. Values are shown as mean ± SD, with significance assessed by a two-tailed unpaired *t* test.

phase in CK-666-treated U2OS cells, which might contribute to the effect. While we employed a quantified EdU incorporation assay with DAPI staining of DNA amount, Schrank et al presented only a non-quantified DAPI staining, which might not be sensitive enough to detect smaller alterations of the cell cycle. Another reason could be the cell line drift due to genomic instability of cancer cell lines (Ben-David et al, 2018; Quevedo et al, 2020). In a previous study, Muqing Cao et al showed that treatment with CK-666 led to cell cycle arrest in RPE1 cells, but this effect was abrogated by RB pathway suppression (Cao et al, 2023). They also observed that CK-666 induced cell cycle arrest in only one out of five tested cancer cell lines, suggesting that oncogene activation may enable some cancer cells to evade CK-666-induced cell cycle arrest. Therefore, although the same U2OS cell line was used, variations in the source

of the cell lines could potentially lead to genetic differences, resulting in varying effects of CK-666 on cell cycle arrest.

Nieminuszczy et al conducted proximity ligation assays (PLA) to demonstrate the localization of WASP at ligation forks of HeLa cells (Nieminuszczy et al, 2023). However, HeLa cells were derived from cervical cancer and are therefore not expected to express WASP. Of note, siRNA-mediated knockdown of WASP in these cells was only demonstrated on mRNA level, not on protein level. In addition, Nieminuszczy et al use the same problematic WASP antibody as Schrank et al, which will result in strong non-specific nuclear binding in the PLA. The authors of this study proposed that WASP and N-WASP-dependent nuclear actin polymerization is crucial for releasing RPA for binding to single-stranded DNA, which should also be crucial for efficient HDR. In our HDR assays using knock-out of either

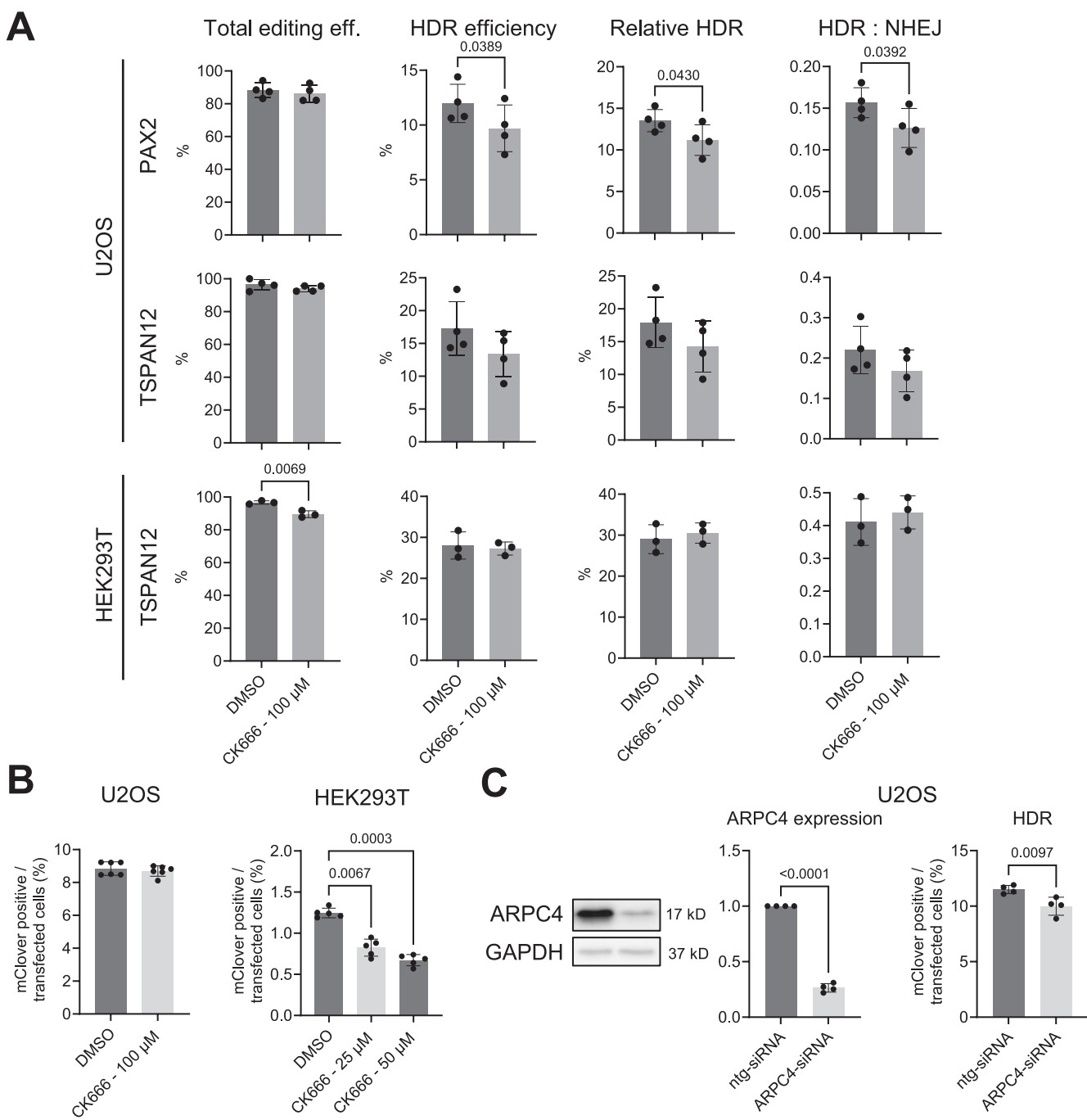

**Figure 10. Arp2/3 inhibition shows cell type and target gene-specific effect on HDR efficiency.**

Indicated cell types pre-treated for 1 h in CK666 or DMSO were transfected with plasmids encoding Cas9 and an sgRNA targeting indicated endogenous genes (TSPAN12, PAX2) and corresponding single-stranded repair templates for IDAA experiments, or with Cas9, an sgRNA for LMNA, and a double-stranded repair plasmid inserting the fluorescent mClover. (A) Total editing, HDR efficiency, and HDR:NHEJ ratio as measured by IDAA. Each dot represents one independent experiment. (B) HR-mediated insertion of mClover in indicated cell types. Each dot represents one independent experiment. (C) Effective knockdown of ARPC4 in U2OS cells by siRNA shown by Immunoblot analysis, resulting in significantly reduced HR repair as measured by mClover assay. Values are shown as mean ± SD, with significance assessed by a two-tailed unpaired *t* test or one-way ANOVA test with Turkey correction for multiple comparisons. Source data are available online for this figure.

endogenous N-WASP or WASP and overexpression of HA-tagged forms of the proteins, however, we could not see such an effect.

Han et al showed by PLA in lymphoid cells that WASP associated with γH2AX, RPA1, RPA2, and ARP2 in response to replication stress (Han et al, 2022). Whether the problematic WASP antibody was used for these PLA experiments was not stated. They further report that WASP binding to RPA promotes RPA association with ssDNA. By overexpression of WASP in WASP knock-out U2OS cells, however, we could not observe an increased efficiency of HDR, which is dependent on RPA.

Drosophila has only one gene orthologue to WASP and N-WASP. A previous study investigating the function of actin nucleators in DSB repair in Drosophila showed that depletion of Scar and Wash, but not of Wasp, impaired Arp2/3-dependent relocalization of heterochromatic DSB, a process which facilitates HDR (Caridi et al, 2018). These data are in line with a less important function of WASP and N-WASP in HDR.

Tagging of N-WASP and WASP with HA as used in our study has the advantage of efficient detection of the tagged molecules by immunofluorescent staining, but the potential risk of reduced functionality because of steric hindrance. Yet, neither overexpression of a tagged or a non-tagged WASP changed HDR efficiency of U2OS cells. Furthermore, we showed earlier that HA-tagging of N-WASP did not prevent nuclear localization, actin polymerization, or nuclear colocalization with WIP, a protein binding to the N-terminal domain of N-WASP adjacent to the HA-tag (Jiang et al, 2025).

Gene knock-out induced by CRISPR/Cas as employed in this study could lead to off-target mutations and corresponding phenotypes. However, it is extremely unlikely that an off-target effect completely rescues the phenotype of an on-target mutation. Furthermore, we used polyclonal knock-out cells, reducing the influence of any infrequent mutation event.

Studying the function of nuclear actin polymerization in DNA repair is difficult, since it is not possible to specifically interfere with nuclear actin polymerization. Manipulating the amount of polymerizable actin in the nucleus might affect the stability of chromatin remodeling complexes that bind to monomeric actin, such as INO80 and SWR (Willhoft and Wigley, 2020). In addition, interfering with actin polymerization might result in indirect effects caused by altered cytoplasmic actin polymerization. Investigating nuclear actin polymerization by nucleus-targeted, actin-binding fluorescent molecules might stabilize F-actin, resulting in artificial nuclear actin polymerization (Belyy et al, 2020; Flores et al, 2019; Jiang et al, 2025).

Investigating DNA repair by cell cycle-resolved γH2AX kinetic has the advantage that besides the induction of DNA damage no further manipulation of the living cells is required. Besides the quantification of γH2AX foci per nucleus, we think it is important to check in addition the mean fluorescence of the pixels in the nucleus, since alterations in size or intensity of the γH2AX foci might affect their recognition by the image analysis programs. Mean intensity measurement does not require foci identification and therefore avoids any corresponding problems. DNA repair assay relying on transfection might be influenced indirectly, if the mechanism to be studied is affecting transfection efficiency. It is possible that such effects contributed to cell-type-specific differences in HDR efficiency in the response to the Arp2/3 inhibitor CK-666.

Very recently, Woodward et al demonstrated that the formin DIAPH1 regulates DNA repair via γ-actin and identified natural human mutations of DIAPH1 associated with defective DSB repair (Woodward et al, 2025). They also report a role for the Arp2/3 complex, but its interplay with DIAPH1 in DSB repair is not clear.

While it is proven that actin polymerization can occur in the nucleus and that the nucleus contains many actin-binding molecules and nucleation-promoting factors, the data provided in this study indicate that their role in DNA repair is still far from understood.

# Methods

### Reagents and tools table

| Reagent/resource | Reference of source | Identifier or catalog number |
|---|---|---|
| **Experimental models** | | |
| HEK293T cells (*H. sapiens*) | ATCC | CRL-3216 |
| U2OS cells (*H. sapiens*) | ATCC | HTB-96 |
| MCF-10A cells (*H. sapiens*) | ATCC | CRL-10317 |
| **Recombinant DNA** | | |
| lentiCRISPRv2 | Addgene | #52961 |
| lentiCRISPRv2-WAS | This article | |
| lentiCRISPRv2-WASL | This article | |
| pCAG-GFP | Addgene | #11150 |
| pCAG-WAS-IRES-GFP | This article | |
| pCMV-VSV-G | Addgene | #8454 |
| pCR2.1-mClover-LMNA-Donor | Addgene | #122508 |
| piRFP670-N1 | Addgene | #45457 |
| pLX330-LMNA-gRNA-1 | Addgene | #122507 |
| pRRL MND WASp | Addgene | #36248 |
| pRRLSIN.cPPT.PGK-GFP.WPRE | Addgene | #12252 |
| pRRLSIN.cPPT.PGK-WAS-IRES-BSD | This article | |
| pRRLSIN.cPPT.PGK-WASL-IRES-BSD | This article | |
| psPAX2 | | #12260 |
| pSpCas9(BB)-2A-Puro (PX459) V2.0 | Addgene | #62988 |
| **Antibodies** | | |
| ARPC4 | Abcam | ab217065 |
| GAPDH | Sigma-Aldrich | G9545 |
| Goat Anti-Mouse IgG Antibody (H + L) | Vector Laboratories | PI-2000 |
| Goat Anti-Rabbit IgG Antibody (H + L) | Vector Laboratories | PI-1000 |
| Goat Anti-Mouse IgG (H + L) Cross-Adsorbed Secondary Antibody, Alexa Fluor™ 488 | Invitrogen | A-11001 |

| Reagent/resource | Reference of source | Identifier or catalog number |
|---|---|---|
| Goat Anti-Mouse IgG (H + L) Cross-Adsorbed Secondary Antibody, Alexa Fluor™ 647 | Invitrogen | A-21235 |
| Goat Anti-Rabbit IgG (H + L) Cross-Adsorbed Secondary Antibody, Alexa Fluor™ 488 | Invitrogen | A-11008 |
| Goat Anti-Rabbit IgG (H + L) Cross-Adsorbed Secondary Antibody, Alexa Fluor™ 568 | Invitrogen | A-11011 |
| HA-Tag (C29F4) | Cell Signaling | 3724 |
| N-WASP (30D10) | Cell Signaling | 4848 |
| Vinculin | Sigma-Aldrich | V9131 |
| WASP (D-1) | Santa Cruz | sc-5300 |
| **Oligonucleotides and other sequence-based reagents** | | |
| **Single guide RNAs** | | **Sequence (5′–3′)** |
| Non-targeting gRNA | Integrated DNA Technologies, Inc. | GCTTAGTTACGCG TGGACGA |
| PAX2 | Integrated DNA Technologies, Inc. | GCAAAGTGGCGAC GCCCAAAG |
| TSPAN12 | Integrated DNA Technologies, Inc. | GATGTTAGGATA TTGTGGAA |
| WAS | Integrated DNA Technologies, Inc. | GGTATGTTCTGCT GAACCGC |
| WASL-68 | Integrated DNA Technologies, Inc. | GAGGGACTCGTT CTCCTGCG |
| WASL-157 | Integrated DNA Technologies, Inc. | TGCAGTTATATG CAGCAGAT |
| WASL-169 | Integrated DNA Technologies, Inc. | CAGCAGATCGGA ACTGTATG |
| **siRNAs** | | |
| ON-TARGETplus Human ARPC4 siRNA | Dharmacon | L-008571-00-0005 |
| ON-TARGETplus Non-targeting Control siRNA | Dharmacon | D-001810-10 |
| **Primers for IDAA** | | **Sequence (5′–3′)** |
| 6-FAM-universal | Integrated DNA Technologies, Inc. | 6-FAM-AGCTGACC GGCAGCAAAATTG |
| hPAX2-F1 | Integrated DNA Technologies, Inc. | GAGGGATGGTACC CCTTGTCC |
| hPAX2-R1 | Integrated DNA Technologies, Inc. | AGCTGACCGGCAG CAAAATTGAGGAGCC GGTCTCGAATCTC |
| hPAX2-F2 | Integrated DNA Technologies, Inc. | AGCTGACCGGCAGC AAAATTGCCTATTT GGGCTGGTGCAA |
| hPAX2-R2 | Integrated DNA Technologies, Inc. | CCTGTTGATGGA AGAGACGCT |
| hTSPAN12-F | Integrated DNA Technologies, Inc. | AGCTGACCGGCAGC AAAATTGTGCAGTCAC TAGTTTTGAGGCAT |

| Reagent/resource | Reference of source | Identifier or catalog number |
|---|---|---|
| hTSPAN12-R | Integrated DNA Technologies, Inc. | ACGAAAGCGTCC CTTCTTACA |
| **Chemicals, enzymes, and other reagents** | | |
| Normal goat serum | Cell Signaling | 5425 |
| Benzonase nuclease | Millipore | E1014 |
| BglII | Thermo Scientific | ER0081 |
| Blasticidin S HCl | Gibco | A1113903 |
| BpiI (BbsI) | Thermo Scientific | ER1012 |
| BSA fraction V | PanReac AppliChem | A1391 |
| Camptothecin | MedChemExpress | HY-16560 |
| Cholera toxin | Sigma-Aldrich | C8052 |
| CK-666 | Abcam | ab141231 |
| Click-iT™ EdU Cell Proliferation Kit for Imaging, Alexa Fluor™ 647 dye | Invitrogen | C10340 |
| Click-iT™ Plus EdU Alexa Fluor™ 647 Flow Cytometry Assay Kit | Invitrogen | C10634 |
| Click-iT™ Plus Alexa Fluor™ 488 Picolyl Azide Toolkit | Invitrogen | C10641 |
| cOmplete Protease Inhibitor Cocktail | Roche | 11873580001 |
| DAPI | Thermo Scientific | 62248 |
| DMEM/F-12, HEPES | Gibco | 31330038 |
| DMEM, high glucose, GlutaMAX™ Supplement | Gibco | 61965059 |
| DMSO | Merck | D2650 |
| Esp3I (BsmBI) | Thermo Scientific | ER0451 |
| FastDigest BshTI | Thermo Scientific | FD1464 |
| FastDigest SalI | Thermo Scientific | FD0644 |
| Fetal bovine serum | Cytiva HyClone | SV30160.03 |
| Formaldehyde 4% aqueous solution, buffered | VWR Chemicals | 9713 |
| GenJet™ In Vitro DNA Transfection Reagent optimized for U2OS cells | SignaGen | SL100489-OS |
| Horse serum, New Zealand origin | Gibco | 16050122 |
| HRP substrate | Millipore | WBLUF |
| Human EGF, animal-free recombinant protein | Gibco | AF-100-15 |
| Hydrocortisone | Sigma-Aldrich | H0888 |
| Insulin from bovine pancreas | Sigma-Aldrich | I1882 |
| Mounting media | Agilent Dako | S3023 |
| NEBuilder HiFi DNA Assembly Master Mix | New England Biolabs | E2621 |

| Reagent/resource | Reference of source | Identifier or catalog number |
|---|---|---|
| NuPAGE LDS Sample Buffer | Invitrogen | NP0008 |
| NuPAGE Sample Reducing Agent | Invitrogen | NP0009 |
| Oligofectamine Transfection Reagent | Invitrogen | 12252011 |
| Opti-MEM | Gibco | 11058021 |
| Penicillin-Streptomycin | Gibco | 15140-122 |
| Phenylmethylsulfonyl fluoride | Roche | 10837091001 |
| PhosSTOP Phosphatase Inhibitor Cocktail | Roche | 4906837001 |
| Phusion Hot Start II High-Fidelity PCR Master Mixes | Thermo Scientific | F565S |
| Pierce™ BCA Protein Assay Kit | Thermo Scientific | 23225 |
| Poly-D-Lysine | Gibco | A3890401 |
| Polyethylenimine (PEI) | Polysciences, Inc. | 23966 |
| Propidium iodine (PI) | Fluka | 70335 |
| Puromycin | Gibco | A1113803 |
| Q5 High-Fidelity 2X Master Mix | New England Biolabs | M0492L |
| QuickExtract™ DNA Extraction Solution | Lucigen | QE09050 |
| RAD51 Inhibitor B02 | MedChemExpress | HY-101462 |
| RIPA buffer | Sigma-Alrich | R0278 |
| SCR7 pyrazine | MedChemExpress | HY-107845 |
| Skim milk | Sigma-Aldrich | 70166 |
| T4 DNA Ligase | Thermo Scientific | EL0014 |
| Talazoparib | Axon Medchem | 2502 |
| Triton X-100 | VWR Chemicals | 28817.295 |
| Software and analysis tools | | |
| CellProfiler v.4.2.8 | BROAD Institute | https://cellprofiler.org/ |
| CRISPOR | Genomics Institute, University of California at Santa Cruz | https://crispor.gi.ucsc.edu/ |
| DECODR v3.0 | Gene Editing Institute, Helen F. Graham Cancer Center and Research Institute | https://decodr.org/ |
| GraphPad Prism v10.4.2 | Dotmatics | https://www.graphpad.com/ |
| ICE | Synthego | https://ice.editco.bio/#/ |
| Fiji | Max Planck Institute of Molecular Cell Biology and Genetics | https://imagej.net/software/fiji/ |
| FlowJo v10.10.0 | BD Biosciences | https://www.flowjo.com/ |

| Reagent/resource | Reference of source | Identifier or catalog number |
|---|---|---|
| Peak Scanner 2 | Thermo Fisher Scientific | https://resource.thermofisher.com/page/WE28396_2/ |
| Python 3.6 | Python Software Foundation | https://www.python.org/ |

## Cell culture

Human embryonic kidney cell line HEK293T (ATCC, CRL-3216; Manassas, VA, USA) and the human osteosarcoma cell line U2OS (ATCC, HTB-96; Manassas, VA, USA) were grown in high-glucose Dulbecco's modified Eagle's medium with Glutamax supplement (Gibco, 61965059; Grand Island, NY, USA) with 10% fetal bovine serum (Cytiva HyClone, SV30160.03; Logan, UT, USA) and Penicillin (100 U) and Streptomycin (100 µg/mL) (Gibco, 15140-122; Grand Island, NY, USA). Human mammary gland epithelial cell line MCF-10A (ATCC, CRL-10317; Manassas, VA, USA) were grown in DMEM/F-12, HEPES (Gibco, 31330038; Grand Island, NY, USA) supplemented with 5% horse serum (Gibco, 16050122; Grand Island, NY, USA), penicillin (100 U) and Streptomycin (100 µg/mL) (Gibco, 15140-122; Grand Island, NY, USA), 10 µg/mL insulin (Sigma-Alrich, I1882; St. Louis, MO, USA), 0.5 µg/mL hydrocortisone (Sigma-Alrich, H0888; St. Louis, MO, USA), 20 ng/mL EGF (Gibco, AF-100-15; Grand Island, NY, USA), and 100 ng/mL choleratoxin (Sigma-Aldrich, C8052; St. Louis, MO, USA). Cells were kept in a humidified 37 °C incubator with 5% $CO_2$ and split every 2-3 days following the standard protocols.

## Generation of knock-out (KO) and overexpression cell lines

Single-guide RNAs (gRNAs) targeting exon 1 or exon 2 of the coding regions of human N-WASP (*WASL*) and WASP (*WAS*), respectively, were designed using CRISPOR (crispor.tefor.net), with guide sequences selected based on low predicted off-target activity (*WASL-68*: 5'- GAGGGACTCGTTCTCCTGCG-3'; *WASL-157*: 5'-TGCAGTTATATGCAGCAGAT-3'; *WASL-169:* 5'- CAGCA-GATCGGAACTGTATG-3'; *WAS:* 5'-GGTATGTTCTGCTGAAC CGC-3'). A non-targeting gRNA (ntgRNA: 5'- GCTTAGTTACG CGTGGACGA-3') with no predicted binding sites in either the murine or human genome (Suzuki et al, 2016) was used as the control. The DNA oligos encoding the sgRNAs (Integrated DNA Technologies, Inc.; Coralville, IA, USA) were cloned into the lentiviral Cas9-expressing vector lentiCRISPRv2 (a gift from Feng Zhang, Addgene #52961; Watertown, MA, USA) (Sanjana et al, 2014) as previously described (Ran et al, 2013).

The DNA sequences encoding human N-WASP and WASP were amplified from U2OS cDNA and pRRL MND WASp (a gift from David Rawlings, Addgene #36248; Watertown, MA, USA) (Astrakhan et al, 2012), respectively. A bicistronic cassette containing the coding sequence (with or without an N-terminal HA tag), an internal ribosome entry site (IRES), and the blasticidin S deaminase gene was cloned into the pRRLSIN.cPPT.PGK-

GFP.WPRE (a gift from Didier Trono, Addgene #12252; Watertown, MA, USA) between the AgeI and SalI restriction sites using NEBuilder HiFi DNA Assembly Master Mix (New England Biolabs, E2621; Ipswich, MA, USA).

To produce lentiviral particles, lentiCRISPRv2 (or pRRLSIN), pCMV-VSV-G (a gift from Bob Weinberg, Addgene #8454; Watertown, MA, USA) (Stewart et al, 2003), and psPAX2 (a gift from Didier Trono, Addgene #12260; Watertown, MA, USA) were co-transfected to HEK293T at 4:3:3 ratio at the final concentration of 10 µg/mL in Opti-MEM™ (Gibco, 11058021; Grand Island, NY, USA) with 40 µg/mL polyethylenimine (PEI, Polysciences, Inc., #23966; Warrington, PA, USA). The cells were incubated with the transfection mixture overnight, followed by the replacement of fresh complete medium the next day. The virus-containing supernatant was collected 24 h after the medium change to transduce U2OS, HEK293T, and MCF-10A cells. 48 h post-transduction, cells were split and on the following day, selected with 1 µg/mL puromycin (Gibco A1113803; Grand Island, NY, USA) or 6 µg/mL blasticidin S HCl (Gibco A1113903; Grand Island, NY, USA). Knock-out efficiency was confirmed by Sanger sequencing, analyzed by DECODR v3.0 (Bloh et al, 2021) and ICE (Conant et al, 2022), or immunoblotting. Expression of knocked-in tagged-proteins was checked by immunoblotting.

## Gene knock-down

siRNAs were resuspended in 1× siRNA buffer (60 mM KCl, 6 mM HEPES, pH 7.5, 0.17 mM MgCl₂ in DEPC-treated water). One day prior to transfection, U2OS cells were seeded in antibiotic-free medium to reach approximately 30% confluency. Cells were transfected with either an siRNA targeting human ARPC4 (Dharmacon, ON-TARGETplus L-008571-00-0005; Lafayette, CO, USA) or a non-targeting control siRNA (Dharmacon, ON-TARGETplus D-001810-10-05; Lafayette, CO, USA) using Oligofectamine (Invitrogen, 12252011; Carlsbad, CA, USA), following the manufacturer's instructions. The next day, cells were reseeded for cell cycle analysis or homologous recombination assays. Knockdown efficiency was assessed 48 h post-transfection by immunoblotting.

## Immunoblotting

Cells were washed with PBS and lysed in RIPA buffer (Sigma-Alrich, R0278; St. Louis, MO, USA) supplemented with cOmplete Protease Inhibitor Cocktail (Roche, 11873580001; Basel, Switzerland), PhosSTOP Phosphatase Inhibitor Cocktail (Roche, 4906837001; Basel, Switzerland), phenylmethylsulfonyl fluoride (PMSF, Roche, 10837091001; Basel, Switzerland) at 200 µg/mL, and benzonase nuclease (Millipore, E1014; Burlington, MA, USA) at 250 U/mL. Protein concentrations were determined using the Pierce™ BCA Protein Assay Kit (Thermo Scientific, 23225; Waltham, MA, USA). Equal amounts of protein were mixed with NuPAGE LDS Sample Buffer (Invitrogen, NP0008; Carlsbad, CA, USA) and NuPAGE Sample Reducing Agent (Invitrogen, NP0009; Carlsbad, CA, USA), boiled at 70 °C for 10 min and resolved by SDS-PAGE on 4–15% Tris-Glycine gels. Proteins were transferred to PVDF membranes (Millipore, Immobilon, IPFL00010; Burlington, MA, USA) using a wet transfer system. Membranes were then blocked with 5% skim milk (Sigma-Aldrich, 70166; St. Louis, MO,

USA) in TBS-T (Tris-buffered saline with 0.1% Tween-20), followed by overnight incubation at 4 °C with primary antibodies diluted in 5% BSA fraction V (PanReac AppliChem, A1391; Darmstadt, Germany) in TBS-T. The following antibodies were used: N-WASP (Cell Signaling, 4848, 1:1000 dilution; Danvers, MA, USA), WASP (Santa Cruz, sc-5300, 1:500 dilution; Dallas, TX, USA), HA-tag (Cell Signaling, 3724, 1:1000 dilution; Danvers, MA, USA), ARPC4 (Abcam, ab217065; Cambridge, UK), GAPDH (Sigma-Aldrich, G9545, 1:100,000 dilution; St. Louis, MO, USA), Vinculin (Sigma-Aldrich, V9131; St. Louis, MO, USA). After washing, membranes were incubated with anti-rabbit or anti-mouse secondary antibodies (Vector Laboratories, PI-1000 and PI-2000, respectively, 1:10,000 dilution; Newark, CA, USA) for 1 h at room temperature. Protein bands were detected using HRP substrate (Millipore, WBLUF; Burlington, MA, USA) and visualized with the ChemiDoc MP Imaging System (Bio-Rad Laboratories; Hercules, CA, USA). The results were quantified using Fiji (Schindelin et al, 2012) and normalized to the expression of GAPDH or Vinculin on the same membrane.

## Wound-healing assay

One day before the experiment, U2OS cells were seeded into 96-well plates to reach 100% confluency. Six hours before the wound-scratching, the media were changed to serum-free to stop cell proliferation. The wounds were created using the WoundMaker™ (Essen BioScience, Sartorius AG; Ann Arbor, MI, USA), and the floating cells were washed off with PBS, and serum-free were added. Wound images were acquired every 4 h using the IncuCyte ZOOM (Essen BioScience, Sartorius AG; Ann Arbor, MI, USA) with a ×10 air objective. Cell migration was evaluated by the percentages of wound area that was closed (wound-closure) after 0, 4, 8, 12, 16, 20, and 24 h.

## DNA double-strand break induction

To induce DNA double-strand breaks, cells were either treated with 100 nM camptothecin in 1 h or irradiated with a Faxitron CP-160 X-ray apparatus (Faxitron Bioptics, LLC., Tucson, AZ, USA). The radiation dose was 2 Gy, delivered at a rate of 1.6 Gy per minute. Control cells received sham irradiation.

## Cell cycle analysis

Cells were seeded either in six-well plates for FACS-based cell cycle analysis or on 13-mm glass coverslips for microscopy-based cell cycle analysis. Seeding densities were adjusted to ensure that cells remained below 80% confluence at the time of sample collection. For HEK293T cells, glass coverslips were coated with poly-D-lysine (Gibco, A3890401; Grand Island, NY, USA) following the manufacturer's instructions. For siRNA knockdown conditions, cells were transfected with siRNA one day before seeding. For compound treatment conditions, cells were treated with DMSO (Merck, D2650; Darmstadt, Germany) or CK-666 (Abcam, ab141231; Cambridge, UK) for 24 h prior to incubation with 10 µM EdU (5-ethynyl-2'-deoxyuridine) for 30 min. For untreated conditions, EdU was added the day after seeding.

For the FACS-based analysis, the incorporation of EdU was detected using either the Click-iT™ Plus EdU Alexa Fluor™ 647 Flow

Cytometry Assay Kit (Invitrogen, C10634; Carlsbad, CA, USA), following the manufacturer's instructions, together with overnight DNA staining using 50 µg/mL propidium iodide (Fluka, 70335; Buchs, Switzerland) and 0.5 µg/mL RNase A (Roche, RNASEA-RO; Basel, Switzerland) at 4 °C. The fluorescent signals of cells were detected and recorded using BD LSR Fortessa X20 (BD Biosciences; San Jose, CA, USA), and the FCS files were analyzed using FlowJo v10.10.0 (BD Biosciences; Ashland, OR, USA).

For the imaging-based analysis the incorporation of EdU was detected using either the Click-iT™ EdU Cell Proliferation Kit for Imaging, Alexa Fluor™ 647 dye or the Click-iT™ Plus Alexa Fluor™ 488 Picolyl Azide Toolkit (Invitrogen, C10634, C10340, and C10641 respectively; Carlsbad, CA, USA), following the manufacturer's instructions, together with DNA staining using 1 µg/mL DAPI (Thermo Scientific, 62248; Waltham, MA, USA). Images were acquired using ScanR High-Content Screening Station (Olympus; Tokyo, Japan) with a UPlanSApo 20×/0.75 air objective and analyzed with CellProfiler v.4.2.6 (Stirling et al, 2021) and Python 3.6 (Python Software Foundation; Wilmington, DE, USA) using a code developed by the authors.

## Immunofluorescent staining

Cells seeded on 13-mm-glass coverslips were wash with PBS and pre-extracted using PBS containing 0.1% Triton X-100 (VWR Chemicals, 28817.295; Radnor, Pennsylvania, United States) for 30 s if needed or proceed directly to fixation using buffered formaldehyde 4% aqueous solution (VWR Chemicals, 9713; Radnor, Pennsylvania, USA) for 15 min at room temperature, followed by permeabilization and blocking in PBS containing 5% normal goat serum (Cell Signaling, 5425) and 0.3% Triton X-100 for 1 h. Cells were then processed for EdU detection, if required, or incubated directly overnight at 4 °C with primary antibodies diluted in PBS containing 1% BSA and 0.3% Triton X-100. After washing, cells were incubated with species-appropriate secondary antibodies (goat anti-mouse IgG (H + L) cross-adsorbed secondary antibody, Alexa Fluor 488 or 647; goat anti-rabbit IgG (H + L) cross-adsorbed secondary antibody, Alexa Fluor 488 or 568; Invitrogen, A-11001, A-21235, A-11008, A-11011, respectively, all with 1:1000 dilution; Carlsbad, CA, USA) for 1 h at room temperature. Filter sets were selected to prevent spectral bleed-through, and the absence of signal leakage was confirmed using negative controls, such as knock-out cell lines or untreated samples lacking the target antigen. Nuclei were counterstained with DAPI (Thermo Scientific, 62248; Waltham, MA, USA) at 1 µg/mL for 5 min, followed by three washes with PBS. Coverslips were mounted with 5 µL of mounting medium (Agilent Dako, S3023; Santa Clara, CA, USA) and imaged using either a ScanR High-Content Screening Station (Olympus; Tokyo, Japan) with a UPlanSApo 20×/0.75 air objective or a ZEISS LSM 800 confocal microscope with a Plan-Apochromat 60×/1.4 oil objective (all from Carl Zeiss Microscopy GmbH; Oberkochen, Germany), depending on the experiment.

## Image analysis

For the wound-healing assay, images were first converted to 8-bit format, and specific spatial frequencies were enhanced using a bandpass filter in Fiji (Schindelin et al, 2012). Subsequently, the images were analyzed in Python using the scikit-image package (Van

der Walt et al, 2014). The wound regions were segmented using a global entropy-based threshold calculated from images at the 0-h timepoint to avoid threshold drift at later timepoints, followed by Otsu thresholding, and wound areas were then quantified from the resulting binary masks with a monotonic rule. Images with unclear or undefinable wound borders were removed from the analysis.

Immunofluorescent image analysis was performed using Cell-Profiler software version 4.2.8 (Stirling et al, 2021). Nuclei were identified using DAPI staining, and cell areas were delineated by the watershed of HA images or by expanding a specified distance from the nuclei. Background signals were removed by subtracting the mean intensity of non-signal areas. All the intensity measurements were performed on the original images after background subtraction. Nuclei boundaries were determined based on intensity gradients through the minimal entropy segmentation algorithm. Nuclei or cells located at the image borders were excluded from the analysis. In CPT-induced DSB experiments, cells with DNA damage were identified by comparing their nuclear γH2AX intensities to those of DMSO-treated controls. Cells with nuclear γH2AX intensities exceeding the 95th percentile of the DMSO-treated control distribution were classified as damaged. γH2AX staining images were enhanced using the "EnhanceOrSuppressFeatures" module with "Speckles" feature option. The enhanced images were then used for γH2AX foci detection using the Otsu method with three-class thresholding with the pixels in the middle intensity class assigned to background. S-phase cells were recognized by the presence of nuclear EdU signals. Colocalization was determined using the MeasureColocalization function of CellProfiler.

## Indel detection by amplicon analysis (IDAA)

IDAA was performed as previously described (Bischoff et al, 2023) with minor modifications. Cells were transfected with a mixture of the pSpCas9(BB)-2A-Puro (PX459) V2.0 plasmid (Addgene #62988; Watertown, MA, USA), which encodes a single guide RNA (sgRNA) targeting an endogenous gene (either *PAX2*: 5'-GCAAAGTGGCGACGCCCAAAG-3', or *TSPAN12*: 5'-GATGT-TAGGATATTGTGGAA-3'), and a single-stranded oligodeoxynucleotide (ssODN) donor template. The donor DNA was 108 nucleotides long, consisting of two 50-nt homology arms flanking an 8-nt asymmetric insertion. The plasmid and donor template were transfected at a 2:1 mass ratio using GenJet™ In Vitro DNA Transfection Reagent optimized for U2OS cells (SignaGen, SL100489-OS; Frederick, MD, USA) following the manufacturer's instructions, or using polyethylenimine (PEI; Polysciences, Inc., 23966; Warrington, PA, USA) at a ratio of 4 µL PEI per 1 µg total DNA for $5 \times 10^5$ HEK293T cells. Treatment with DMSO or 100 µM CK-666 (Abcam, ab141231; Cambridge, UK) started 1 h before transfection and continued till harvesting. New media were changed 6 h post transfection.

Forty-eight to seventy-two hours post-transfection, cells were harvested and co-stained with 1 µg/mL 7-AAD (Invitrogen, A1310, Carlsbad, CA, USA). Successfully transfected cells were isolated by fluorescence-activated cell sorting (FACS) using a BD FACSMelody™ Cell Sorter (BD Biosciences, San Jose, CA, USA). An untransfected sample was used as a negative control, in which all GFP-A values fell below a threshold of $10^2$ on a logarithmic scale. In total, 30,0000 cells with GFP-A values above $10^3$ on the logarithmic scale per sample were collected for genomic DNA

isolation. The edited loci were amplified from the genomic DNA using a fluorescent tri-primer PCR method, which employed a gene-specific primer pair—one of which was tailed with a universal sequence—along with a fluorescently labeled universal primer (6-FAM-AGCTGACCGGCAGCAAAATTG) for detection. This approach enables precise sizing of amplicons by fragment length analysis. The primers used in for IDAA include: hPAX2-F1, 5'-GGAGGGATGGTACCCCTTGTCC-3'; hPAX2-R1, 5'-AGCTGAC CGGCAGCAAAATTGAGGAGCCGGTCTCGAATCTC-3; hPAX 2-F2, AGCTGACCGGCAGCAAAATTGCCTATTTGGGCTGGTG-CAA; hPAX2-R2, CCTGTTGATGGAAGAGACGCT; hTSPAN12-F, 5'- AGCTGACCGGCAGCAAAATTGTGCAGTCACTAGTTTTGA GGCAT-3'; hTSPAN12-R, 5'-ACGAAAGCGTCCCTTCTTACA-3'. The universal sequence is italicized and underlined.

Fragment length analysis was performed by GENEWIZ Azenta (South Plainfield, NJ, USA). Peaks from .fsa files were identified using Peak Scanner 2 software (Thermo Fisher Scientific, Waltham, MA, USA), and editing outcomes—including total editing, homology-directed repair (HDR), and non-homologous end joining (NHEJ)—were quantified based on corresponding peak areas. For peak detection, a threshold of 100 relative fluorescent units was applied to eliminate background noise. The minimum peak half-width was set to 1, with a polynomial degree of 3 and a peak window size of 7. Peak slope start and stop values were both set to 0. Baseline window sizes ranged from 21 to 51, depending on peak quality, with consistent settings applied across all samples within the same experimental set. No peak smoothing was used, and the Local Southern method (Southern, 1979) was employed for size calling. Peaks within ±25 bp of the wild-type peak were included in the analysis.

## Genomic DNA extraction and genomic PCR

Cell pellets were resuspended with QuickExtract™ DNA Extraction Solution (Lucigen, QE09050, Middleton, WI, USA) at a ratio of 1000 cells per 1 µL of reagent. Cell suspensions were heated at 65 °C for 15 min and then at 98 °C for 5 min to extract genomic DNA. Genomic PCR was performed using Phusion Hot Start II High-Fidelity PCR Master Mixes (Thermo Scientific, F565S, Waltham, MA, USA).

## Homologous recombination assay

The pLX330-LMNA-gRNA-1 plasmid, which encodes a single-guide RNA (sgRNA) targeting the LMNA locus; the pCR2.1-mClover-LMNA-Donor plasmid (a gift from Graham Dellaire, Addgene, #122507 and #122508, respectively; Watertown, MA, USA) (Pinder et al, 2015), which served as the double-stranded donor template; and the piRFP670-N1 plasmid (a gift from Vladislav Verkhusha, Addgene #45457; Watertown, MA, USA) (Shcherbakova and Verkhusha, 2013), which served as the transfection control, were mixed at a 1:1:0.1 mass ratio. The plasmid mixture was transfected using GenJet™ In Vitro DNA Transfection Reagent optimized for U2OS cells (SignaGen, SL100489-OS; Frederick, MD, USA), following the manufacturer's instructions, or using polyethylenimine (PEI; Polysciences, Inc., 23966; Warrington, PA, USA) at a ratio of 4 µL PEI per 1 µg total DNA for $5 \times 10^5$ HEK293T cells.

Treatment with DMSO or 100 µM CK-666 (Abcam, ab141231, Cambridge, UK) was initiated 1 h before transfection and maintained until cell harvesting. siRNA knockdown was performed

2 days prior to transfection. Fresh medium was replaced 6 h after transfection.

Three days after transfection, cells were harvested and fixed with 4% aqueous formaldehyde (VWR Chemicals, 9713, Radnor, PA, USA) for 15 min and analyzed by flow cytometry using a BD LSR Fortessa X20 flow cytometer (BD Biosciences; San Jose, CA, USA). HDR efficiency was calculated as the proportion of mClover-positive cells among iRFP670-positive (transfected) cells.

## Flowcytometry analysis

Flow cytometry standard (.fcs) files were analyzed using FlowJo v10.10.0 (BD Biosciences; Ashland, OR, USA). Cells were first gated to identify the intact cell population using forward scatter area (FSC-A) versus side scatter area (SSC-A) plots. Doublets were excluded by gating on forward scatter area (FSC-A) versus forward scatter height (FSC-H) plots. For IDAA experiments, dead cells were subsequently excluded by gating single cells based on 7-AAD fluorescence (7-AAD-A) versus FSC-A. Transfected cells were identified by GFP fluorescence using GFP-A versus FSC-A plots defined as the percentage of cells with GFP-A values above $10^2$ on a logarithmic scale based on a negative control. For the homologous recombination assay, single cells were gated to identify transfected cells based on iRFP670 fluorescence versus FSC-A. HDR-positive cells within the transfected population were identified by the presence of mClover fluorescence using mClover versus FSC-A plots. Positivity thresholds were defined using one double-negative control (negative for both mClover and iRFP670) and two single-positive controls (positive for either mClover or iRFP670 alone).

## Cell competitive growth assay

KO and WT cells were mixed at a 1:1 ratio and cultured under standard conditions with the presence of 0.1% DMSO or 10 nM talazoparib (Axon Medchem, 2502, Groningen, Netherlands). Genomic DNA was extracted at day 0 and day 10 to PCR-amplify the genomic region flanking the target exon for Sanger sequencing (Eurofins Genomics). The .ab1 trace files were then analyzed using ICE (Conant et al, 2022) to determine the KO efficiency. The relative fitness of knockout (KO) cells compared with wild-type (WT) cells under either normal conditions or PARP inhibitor-induced synthetic lethality was assessed by normalizing the KO efficiency after 10 days of co-culture to the initial KO efficiency at day 0.

## Statistical analysis

Statistical analysis was performed using GraphPad Prism v10.4.2 (Dotmatics, San Diego, CA, USA). Data are expressed as mean ± standard deviation. For comparisons between two groups, a two-tailed Student's *t* test was applied. In experiments involving more than two groups, one-way ANOVA followed by Tukey's post hoc test was used to assess significance. Statistical differences are indicated by asterisks as follows: not significant (ns), $P \geq 0.05$; $*P < 0.05$; $**P < 0.01$; $***P < 0.001$; $****P < 0.0001$.

# Data availability

Source data are available at the BioStudies BioImage Archive (S-BIAD2513).

The source data of this paper are collected in the following database record: biostudies:S-SCDT-10_1038-S44319-026-00771-y.

## Peer review information

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

## Acknowledgements

We thank Volkan Turan and Allan Pedersen Zacchi for their technical support. In addition, we thank Dr. Xin Jiang and Dr. Yayashree Vijay Thatte for guidance on experimental procedures, core facility expert Dr. Yasuko Antoku for assistance with microscopy; core facility expert Dr. Rajesh Somasundaram for support with the use of the FACS Melody system; and Dr. Amalie Engstrøm for instruction on the use of the Incucyte. This project has received funding from the European Union's Horizon 2020 research and innovation program under the Marie Skłodowska-Curie grant agreement No 101034291 and from the Danish Cancer Society (Grant R302-A17455).

## Author contributions

**Thu Han Le Phan**: Conceptualization; Data curation; Formal analysis; Investigation; Methodology; Writing—original draft; Writing—review and editing. **Anders Buchard**: Methodology. **Cord Brakebusch**: Conceptualization; Supervision; Funding acquisition; Writing—original draft; Project administration; Writing—review and editing.

Source data underlying figure panels in this paper may have individual authorship assigned. Where available, figure panel/source data authorship is listed in the following database record: biostudies:S-SCDT-10_1038-S44319-026-00771-y.

## Disclosure and competing interests statement

The authors declare no competing interests.

# Expanded View Figures

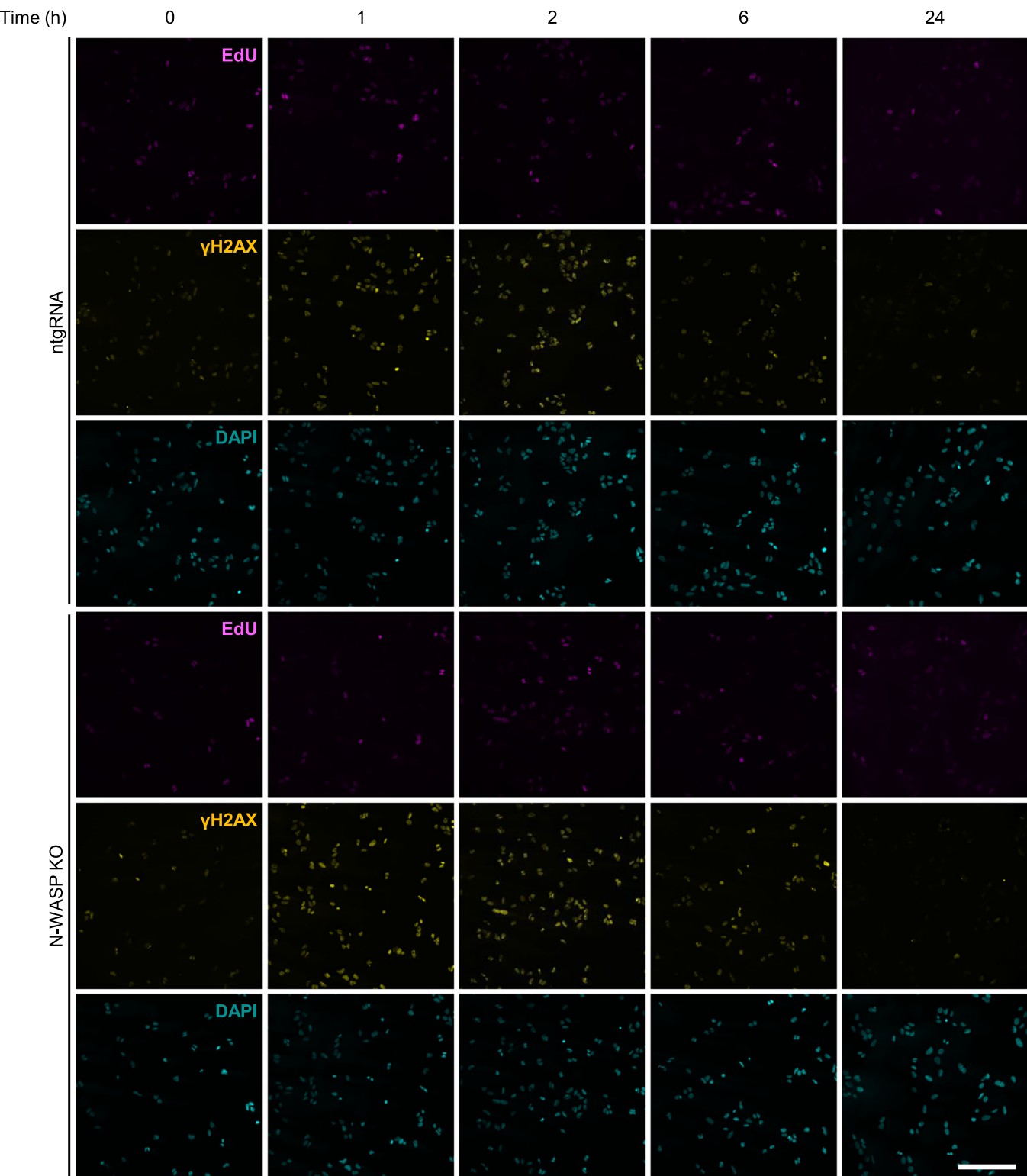

**Figure EV1.  Loss of N-WASP is not increasing gH2AX foci after irradiation.**

Representative images of U2OS' γH2AX and EdU stainings at the indicated time points following exposure to 2 Gy X-ray irradiation; Sham-irradiated cells served as the 0 h controls. Cells were incubated with 10 µM EdU for 30 min prior to collection. Scale bar: 200 µM.

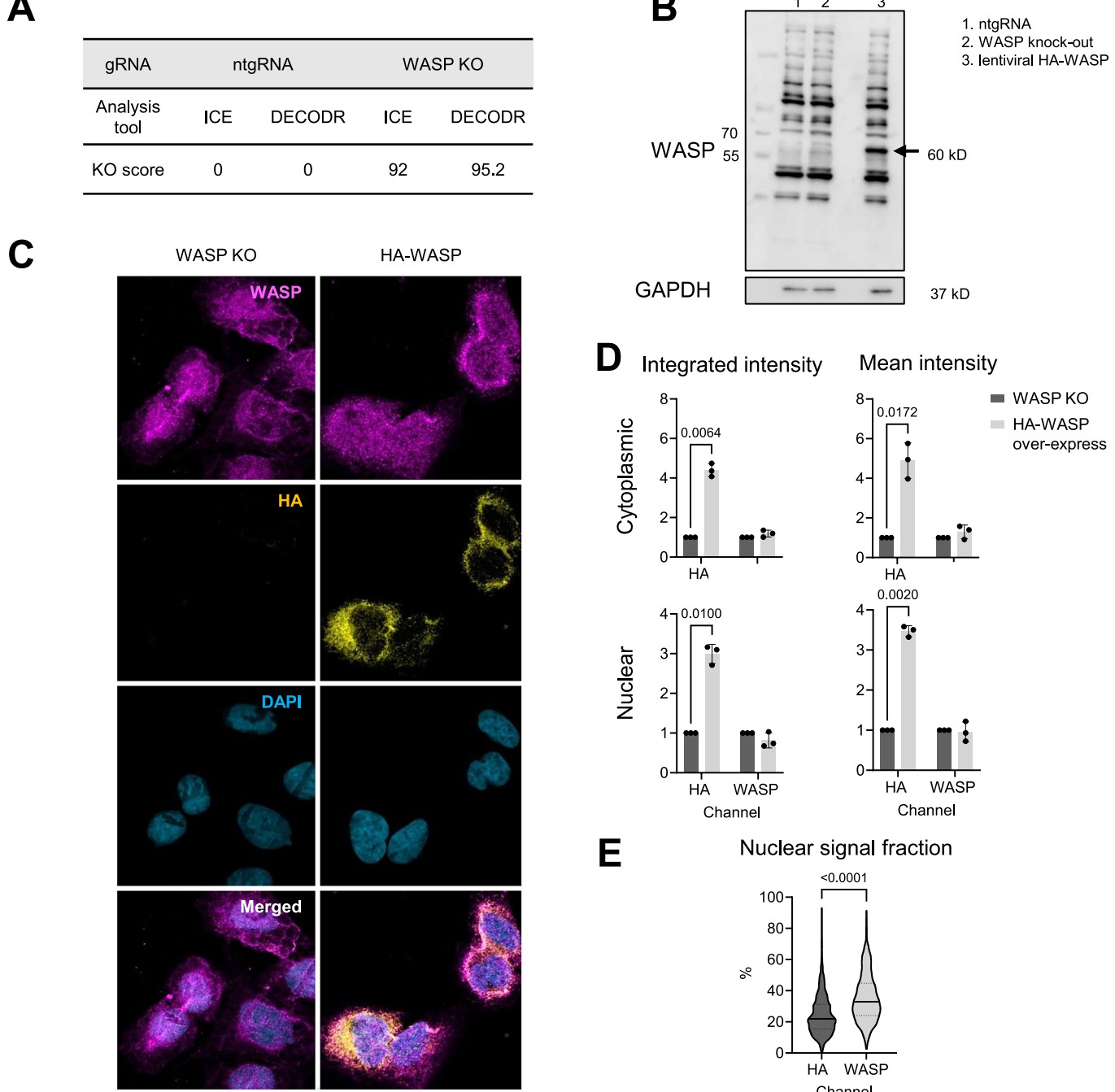

**Figure EV2. WASP antibody validation.**

(**A**) Assessment of WASP knock-out efficiency in U2OS cells through Sanger sequencing of genomic PCR products, followed by analysis using DECODR and ICE. (**B**) Immunoblotting of control transduced (ntgRNA), WASP knock-out, and HA-WASP-overexpressing (lentiviral HA-WASP) U2OS cell lines. (**C**) Immunofluorescence staining of WASP knock-out U2OS cells with or without overexpression of HA-tagged WASP via lentiviral vector. Scale bar: 20 µm. (**D**) Quantification of HA and WASP signal intensities in the two groups from (**C**), normalized to the WASP knock-out (W-KO) group. Each dot represents the average value from an independent biological replicate, containing 5 images for W-KO and 24–30 images for HA-WASP overexpression. Data are shown as mean ± SD, with statistical significance determined by a two-tailed unpaired *t* test. (**E**) Distribution of HA and WASP signal proportions in nuclear areas of images from the HA-WASP overexpression group, presented as a violin plot with median and interquartile range (lower and upper quartiles), statistical significance was determined by a two-tailed unpaired *t* test.

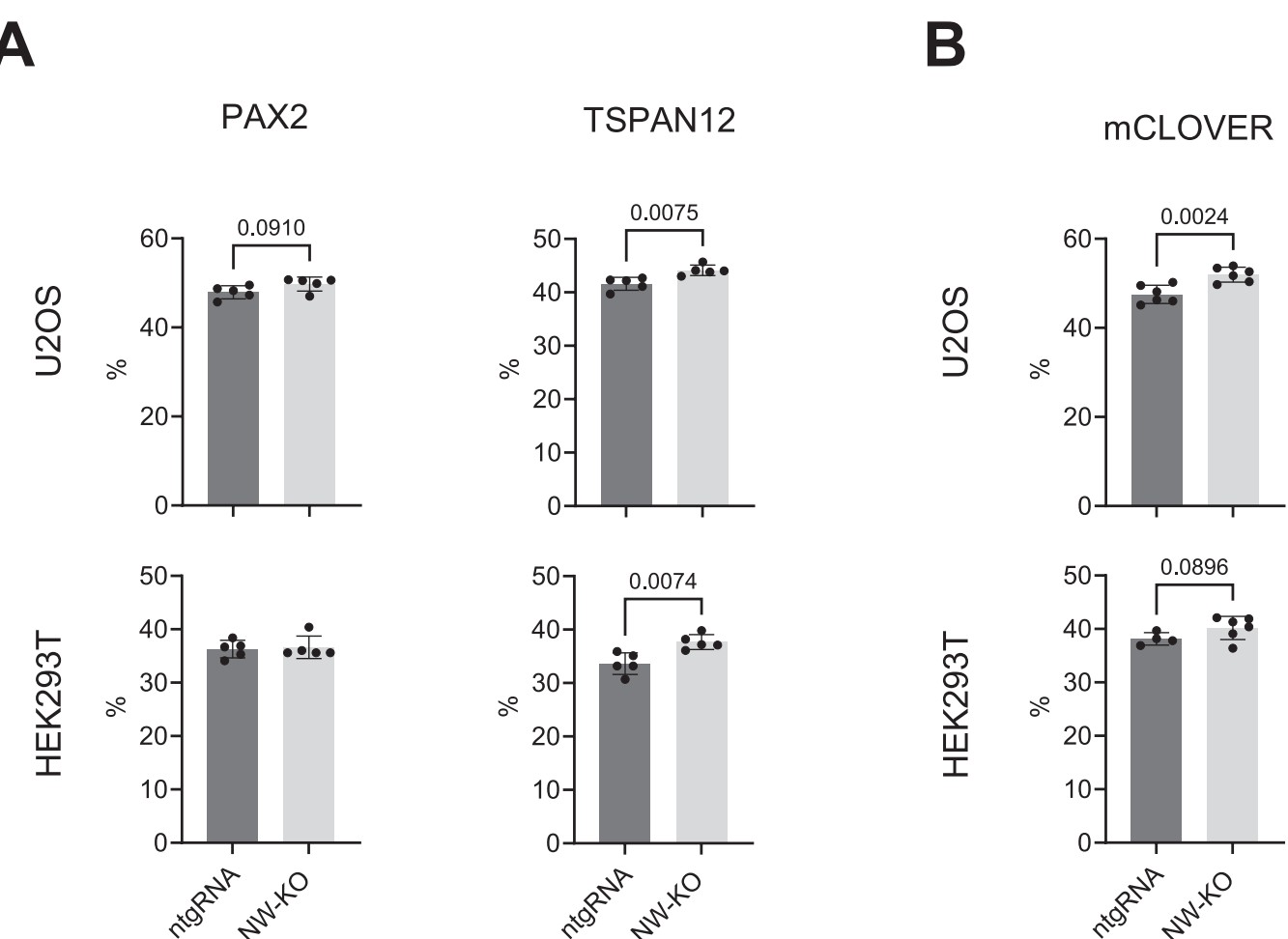

**Figure EV3.   The effects of N-WASP knockout on U2OS cell transfection.**

Indicated cell types with (ntgRNA) and without N-WASP (NW-KO) were transfected with (**A**) plasmids encoding Cas9 and an sgRNA targeting indicated endogenous genes (TSPAN12, PAX2) and corresponding single-stranded repair templates for IDAA experiments, or with (**B**) Cas9, an sgRNA for LMNA, and a double-stranded repair plasmid inserting the fluorescent mClover. Transfection efficiency as determined by FACS for fluorescent transfection marker. Each dot represents an independent experiment. Values are shown as mean ± SD, with significance assessed by a two-tailed unpaired *t* test.

