## [Peer Review File · EMBO Reports]

Dispensable Players: N-WASP and WASP Are Not Crucial for Homology-Directed DNA Repair

Thu Han Le Phan, Anders Buchard, and Cord Brakebusch

Corresponding author(s): Cord Brakebusch (cord.brakebusch@bric.ku.dk)

Review Timeline:

Submission Date:	2nd Jul 25
Editorial Decision:	13th Aug 25
Revision Received:	19th Dec 25
Editorial Decision:	16th Mar 26
Referees' Cross-comments:	16th Mar 26
Revision Received:	19th Mar 26
Accepted:	27th Mar 26

Editor: Esther Schnapp

Transaction Report:

Dear Dr. Brakebusch,

Thank you for the submission of your manuscript to EMBO reports. We have now received the full set of referee reports that is pasted below.

As you will see, while the referees acknowledge that the findings are potentially interesting they also all point out that additional data and evidence is required to support your hypothesis that N-WASP and WASP are not crucial for HDR. I think all points are good and should be addressed but please let me know in case you disagree and we can discuss this further, also in a video chat, if you like.

I would thus like to invite you to revise your manuscript with the understanding that the referee concerns must be fully addressed and their suggestions taken on board. Please address all referee concerns in a complete point-by-point response. Acceptance of the manuscript will depend on a positive outcome of a second round of review. It is EMBO reports policy to allow a single round of major revision only and acceptance or rejection of the manuscript will therefore depend on the completeness of your responses included in the next, final version of the manuscript.

We realize that it is difficult to revise to a specific deadline. In the interest of protecting the conceptual advance provided by the work, we recommend a revision within 3 months (13th Nov 2025). Please discuss the revision progress ahead of this time with the editor if you require more time to complete the revisions.

- 1) A data availability section providing access to data deposited in public databases is missing. If you have not deposited any data, please add a sentence to the data availability section that explains that.
- 2) Your manuscript contains statistics and error bars based on $n=2$. Please use scatter blots in these cases. No statistics should be calculated if $n=2$.

3) We replaced Supplementary Information with Expanded View (EV) Figures and Tables that are collapsible/expandable online. A maximum of 5 EV Figures can be typeset. EV Figures should be cited as 'Figure EV1, Figure EV2' etc... in the text and their respective legends should be included in the main text after the legends of regular figures.

5) a complete author checklist, which you can download from our author guidelines . Please insert information in the checklist that is also reflected in the manuscript. The completed author checklist will also be part of the RPF.

6) Please note that all corresponding authors are required to supply an ORCID ID for their name upon submission of a revised manuscript (. Please find instructions on how to link your ORCID ID to your account in our manuscript tracking system in our Author guidelines

- the name of the statistical test used to generate error bars and P values,
- the number (n) of independent experiments (please specify technical or biological replicates) underlying each data point,
- the nature of the bars and error bars (s.d., s.e.m.),
- If the data are obtained from n {less than or equal to} 2, use scatter blots showing the individual data points.

12) All Materials and Methods need to be described in the main text using our 'Structured Methods' format, which is required for all research articles. According to this format, the Methods section includes a Reagents and Tools Table (listing key reagents, experimental models, software and relevant equipment and including their sources and relevant identifiers) followed by a Methods and Protocols section describing the methods using a step-by-step protocol format. The aim is to facilitate adoption of the methodologies across labs. More information on how to adhere to this format as well as a downloadable template (.docx) for the Reagents and Tools Table can be found in our author guidelines:
<https://www.embopress.org/page/journal/14693178/authorguide#structuredmethods>.

An example of a Method paper with Structured Methods can be found here: <https://www.embopress.org/doi/full/10.1038/s44320-024-00037-6#sec-4>

As part of the EMBO publication's Transparent Editorial Process, EMBO reports publishes online a Review Process File (RPF) to accompany accepted manuscripts. This File will be published in conjunction with your paper and will include the referee

reports, your point-by-point response and all pertinent correspondence relating to the manuscript.

I look forward to seeing a revised form of your manuscript when it is ready.

Referee #1:

This paper from Phan et al. examines the role of N-WASP and WASP in DSB repair in U2OS cells. Previous studies identified a role for WASP in HDR repair (e.g., Schrank, 2018; Ha, 2022; Zigelbaum, 2023). Phan et al. did not find evidence that N-WASP or WASP contribute to this pathway. Specifically, they did not confirm the previously detected colocalization of WASP with repair sites, nor did they detect N-WASP enriched at DSBs. Further, they did not detect HR repair defects upon N-WASP KO or WASP OP measured with an assay for CRISPR editing efficiency. The authors identified effects of CK666 treatment on cell cycle and transfection efficiency, which might indirectly affect HDR, potentially explaining the conclusions of previous studies. Collectively, this study provides evidence against the model that nuclear actin polymerization is required for HDR.

While the topic is important, it is much more difficult to prove the absence of a phenotype than its presence. Lack of phenotypes like colocalization with repair sites could, for example, result from different lots of antibodies or staining conditions. Additional and more rigorous evidence needs to be provided to conclude a lack of effect. For example, colocalizations should be analyzed in Triton-extracted samples, in conditions that highlight nuclear proteins only, and further tested by PLA assays. Effects of CK666 on transfection efficiency and cell cycle (not detected in previous studies) might result from different treatment conditions (time and concentration) and cell growth conditions. Lack of HDR repair responses might result from compensatory effects or less sensitive methods than those used in previous studies. Collectively, while I find the results useful to the scientific community, a comprehensive set of experiments needed to disprove the previous observations is lacking. For example, previous studies ruled out cell cycle effects in arrested cells, used GFP-based reporter assays broadly validated by the community, looked at depletions of different components of the WASP-Arp2/3 axis, and showed recruitment to repair sites by ChIP and other biochemical assays in addition to IF.

This study cannot unequivocally conclude that N-WASP or WASP are dispensable for HDR repair. Given this, I do not recommend its publication in EMBO Reports.

Referee #2:

In this manuscript, the authors investigate the poorly defined role of the actin nucleation regulators N-WASP and WASP in the DNA damage response. The manuscript uses a variety of approaches to carefully examine previously reported phenotypes of actin regulation in DNA repair, and the authors have done well to design controlled experiments. The manuscript thoroughly investigates N-WASP in U2OS cells, characterising cell cycle profiles, cell morphologies and studying DNA repair mechanisms and outcomes. A key component of this work was the use of DNA repair reporter assays which demonstrate the lack of a functional role for either N-WASP or WASP in homologous recombination (HR). This is in contradiction with previous reports, but I believe the authors have robustly designed and carried out their experiments and that the results are reliable, whereas the previous works have flaws that are unresolved in those publications. The manuscript also studies and discusses important points, such as the lack of WASP expression in most cell lines and the reliance on antibodies with low-specificity.

Overall, I believe this work to be of high quality and to address key points of interest within the current literature. Although limited in some aspects, the results presented are important and would make good contributions to the field. I would recommend publication after the following comments are addressed.

Major comments:

1. The authors have focused on the U2OS cell line for comparison with previously published work. I agree with this in principle, however for publication of the findings it would be very beneficial for an additional cell line to be tested that is distinct from U2OS.

Otherwise, there is the potential for the results to be seen as U2OS specific, which I do not believe is the case, and which is not the message of the manuscript. Of course, only a few experiments that the authors deem key to their conclusions need to be confirmed in another cell line, not all experiments.

2. Figure EV2C should have a WT comparison for the signal of the WASP antibody, as if the signal is 10-fold lower in the WASP KO cells then an argument can be made for specificity. Over-expression of the HA-tagged WASP functionally works as a control, and I believe the authors' interpretation of this, but fluorescence intensity is arbitrary and needs sufficient controls for appropriate comparisons. This should also be quantified.

3. γ H2AX foci is only studied in the context of N-WASP KO. It would be very beneficial if WASP KO and possibly even CK-666 treatment was also studied in these experiments for comparison to N-WASP. The previous Schrank et al. manuscript observed delayed DSB repair following CK-666 treatment specifically in G2 cells and only with enzyme induced breaks via qPCR. It would be interesting to compare to this work.

Minor comments:

1. Introduction states RAD51 enables then repair by Homologous Recombination, I believe this should be "then enables repair by" or something similar, I'm not sure if the current form is correct.

2. Figure 1 needs representative images.

3. Page 14 first line, should be tagging of N-WASP and WASP "with HA" not "by HA"

Referee #3:

Brakebusch and colleagues present evidence that in U2OS cells, N-WASP and WASP are not required to mediate HR-dependent repair of DSBs. Given dubious data presented in the highly cited Nature paper by Schrank et al suggesting that the haematopoietic-restricted actin nucleating factor, WASP, plays an important role in regulating DSB repair in epithelial cells, it is important to publish this manuscript questioning these findings. However, since it is not clear to me that the N-WASP KO cell generated in this manuscript is truly null, it is difficult for the authors to make such a strong statement about the role of N-WASP in HR-directed repair. Despite this, with some revisions (detailed below), I believe that this manuscript would be suitable for publication in EMBO Reports.

Specific points:

1. Figure EV1A: the N-WASP CRISPR KO cell line does not look like a complete knockout as there is residual protein. Based on this, it is difficult to conclude whether the lack of a DSB repair, cell migration or replication defect in these cells is due to the presence of residual protein being expressed. Can the residual N-WASP be reduced with siRNA? Does depletion of N-WASP in U2OS cells affect DSB repair or cell migration?

2. Figure 1: 10Gy is a very high dose of ionising radiation. Since it is well established that defects in MRN-dependent processes are better observed following exposure to low doses of IR, it may be that a defect in the resolution of H2AX foci in the N-WASP 'KO' cells is only detectable following exposure to 1-2Gy of IR. However, since IR-induced DSBs are predominantly repaired by NHEJ, it might be better to examine the presence of a DSB repair defect in the N-WASP 'KO' cells using camptothecin, where the DSBs are exclusively repaired by HR.

3. Figure 2: Many proteins involved in the repair of DSBs are not observed as localising to foci using conventional immunofluorescence protocols. This does not mean they are not present at DSBs. It would be more prudent to assess the localisation of N-WASP to DSBs using PLA with γ -H2AX or using the Fok1-induced DSB system generated by the Greenberg lab. This approach has previously been used to show the presence of DIAPH1, another actin nucleating factor, at DSBs (Woodward et al. 2025. Nature Commun. 16:4491).

4. Figure EV2: The inability to detect WASP in U2OS cells by Western blotting is consistent with WASP expression being predominantly limited to haematopoietic cells and puts into question the data demonstrating a role for WASP in promoting DSB repair published in the manuscript by Schrank et al. 2018. This is an important observation that should be published.

5. Figure 3: A positive control should be carried out for the HR and NHEJ assays i.e. either knocking down or inhibiting RAD51 and DNA ligase IV. To control for the cell cycle artifacts induced by CK-666 treatment, using immunofluorescence and co-staining with a marker of the cell cycle e.g. CENPF or Cyclin A/B will allow quantification of HR in S/G2 cells only. Furthermore, knocking down a component of the ARP2/3 complex with siRNA would be a useful control. This has been recently shown by Woodward et al. 2025. (Nature Commun. 16:4491).

6. Figure 4: It was previously demonstrated by Woodward et al. that loss of DIAPH1 gives rise to defective HR, which was more prevalent when assaying it using the CRISPR-based mClover-LMNA HR assay developed by Graham Delaire than using the DR-GFP reporter cell line. This indicates that the type of reporter assay used to measure involvement of the role of the actin nucleation pathway in DSB repair is important i.e. the type of DSB generated may dictate how much inhibiting the actin nucleation pathway affects HR-dependent repair.

7. The recent publication by Woodward et al. demonstrating a role for DIAPH1 and gamma-actin regulating DSB repair should be cited. This manuscript shows that manipulating the ability of cells to nucleate actin through the use of naturally occurring human mutations rather than non-specific chemical inhibitors can affect HR. Moreover, this manuscript is consistent with the authors indicating that other actin nucleating factors, rather than WASP/N-WASP, are important for DSB repair.

Referee #1:

This paper from Phan et al. examines the role of N-WASP and WASP in DSB repair in U2OS cells. Previous studies identified a role for WASP in HDR repair (e.g., Schrank, 2018; Ha, 2022; Zigelbaum, 2023). Phan et al. did not find evidence that N-WASP or WASP contribute to this pathway. Specifically, they did not confirm the previously detected colocalization of WASP with repair sites, nor did they detect N-WASP enriched at DSBs. Further, they did not detect HR repair defects upon N-WASP KO or WASP OP measured with an assay for CRISPR editing efficiency. The authors identified effects of CK666 treatment on cell cycle and transfection efficiency, which might indirectly affect HDR, potentially explaining the conclusions of previous studies. Collectively, this study provides evidence against the model that nuclear actin polymerization is required for HDR.

1. While the topic is important, it is much more difficult to prove the absence of a phenotype than its presence. Lack of phenotypes like colocalization with repair sites could, for example, result from different lots of antibodies or staining conditions. Additional and more rigorous evidence needs to be provided to conclude a lack of effect. For example, colocalizations should be analyzed in Triton-extracted samples, in conditions that highlight nuclear proteins only, and further tested by PLA assays.

We performed IF staining for HA-tagged N-WASP and γ H2AX now with and without Triton-extraction. Surprisingly, while Triton-extraction only slightly reduced nuclear staining for γ H2AX, it completely wiped out nuclear staining for HA-N-WASP, indicating that N-WASP is much less strongly bound to chromatin than γ H2AX (Fig. 4D). For this experiment, we induced DNA damage by camptothecin which is primarily repaired by HDR. Optimizing our IF staining protocol we repeated and extended the colocalization experiments. Following camptothecin treatment, γ H2AX and BRCA1 showed strong partial colocalization in the nucleus (over 95% of Pearson correlation coefficient > 0.5 and over 60% of them > 0.6), while nuclear spots of overexpressed HA-N-WASP or HA-WASP showed no colocalization with γ H2AX (Pearson's correlation coefficient around 0) (Fig. 4C). High specificity of the HA staining was proven by staining of WASP KO and HA-WASP overexpressing U2OS cells, which showed virtually no background in WASP KO cells (Fig. EV3C). High specificity of the γ H2AX staining was demonstrated by the high increase of γ H2AX staining following treatment with irradiation or camptothecin (fx. Fig 4D). These data strongly suggest that WASP or N-WASP are not co-localizing with DNA breaks. Using the optimized antibody binding conditions from the immunofluorescence staining we then conducted a PLA assay using the Duolink kit. However, we unfortunately obtained only background stain. While it is possible to titrate now different antibody concentrations, we decided not to pursue this line, since the data described above argue at least against a strong binding of WASP or N-WASP at DNA double-strand breaks. Furthermore, immunofluorescence was the methodology used earlier by Schrank et al to demonstrate localization of WASP at DSB. We mention in the discussion of the revised manuscript that the presence of a very low amount of N-WASP at the DSBs cannot be excluded.

2. Effects of CK666 on transfection efficiency and cell cycle (not detected in previous studies) might result from different treatment conditions (time and concentration) and cell growth conditions.

We tested now the effect of 24h incubation with 100 μ M CK-666 on the cell cycle using EdU incorporation assay in U2OS, MCF-10A and HEK293T cells. In all cell lines we could observe significant changes in G1, S and G2/M, most striking in HEK 293T cells (Fig. 8A). In addition, siRNA mediated knockdown of ARPC4 in U2OS altered G1 and S phase significantly, although to a lesser extent than CK-666 (Fig. 8B). Following irradiation, CK-666 treatment was significantly increasing G1 after 24h (Fig. 9A), but to a lesser extent than observed in non-irradiated cells. KO of N-WASP, on the other hand, did not affect cell cycle in three different cell lines tested (Fig. 1C).

Interestingly, CK-666 treatment strongly affected the transfection efficiency in U2OS and HEK 293 cells (Fig. 8C), which might affect results of transfection dependent DNA repair assays in a complex manner.

3. Lack of HDR repair responses might result from compensatory effects or less sensitive methods than those used in previous studies.

We performed now cell cycle phase resolved γ H2AX staining kinetics after irradiation in N-WASP KO and control U2OS, MCF-10A and HEK293T cells, but could not detect delayed repair in any of the cells (Fig. 1). We neither observed a significant change in micronuclei after 24h (Fig. 3). Compensation by increased WASP expression in N-WASP KO could be excluded by Western blot analysis.

In U2OS cells, we could not see a decrease of HDR efficiency of repair in Cas9-induced DSB in N-WASP KO, both in a IDAA-based and a reporter-based assay (Fig. 7A, B). Similarly, we could not see a decrease of HDR efficiency in N-WASP KO HEK293T in a IDAA-based assay testing two different target genes (Fig. 7A). However, N-WASP KO HEK293T cells showed a reduced HDR repair in a reporter-based mClover assay (Fig. 7B). Notably, whereas U2OS cells exhibited similar HDR efficiencies across all three tested loci, the absolute HDR efficiency for LMNA in the mClover assay was only around 1%, which is markedly lower than the approximately 25% HDR efficiencies observed for PAX2 and TSPAN12 in the IDAA assay. These cell type specific difference might be influenced by chromatin modifications and differences in the repair mechanisms used in the assay systems (single stranded repair template in IDAA vs double stranded repair template in mClover).

Cells with defective HR repair can show synthetic lethality effects when treated with PARP inhibitor. However, we did not observe a synthetic lethality effect when treating N-WASP KO U2OS or HEK293T cells with the PARP inhibitor Talazoparib (Fig. 7C).

With respect to WASP, we could not detect WASP expression in cells previously used for studies reporting effects of WASP in DNA repair (Fig. 1A) and demonstrated furthermore that strong nuclear background staining of the WASP antibody used might have contributed to the earlier reported results (Fig. EV2, Fig. 5, Fig. 6).

These data strongly argue against an important general function of WASP or N-WASP in DNA repair.

4. Collectively, while I find the results useful to the scientific community, a comprehensive set of experiments needed to disprove the previous observations is lacking. For example, previous studies ruled out cell cycle effects in arrested cells, used GFP-based reporter assays broadly validated by the community, looked at depletions of different components of the WASP-Arp2/3 axis, and showed recruitment to repair sites by CHIP and other biochemical assays in addition to IF. This study cannot unequivocally conclude that N-WASP or WASP are dispensable for HDR repair. Given this, I do not recommend its publication in EMBO Reports.

As described above, we tried in the revised version of the manuscript to address the concerns of the reviewer by providing new experiments and testing additional cell lines. We think that our study is a valuable complementation of the already published investigations on the role of nuclear actin polymerization in DNA repair and that it will be of help for future studies in this complex research area.

Referee #2:

In this manuscript, the authors investigate the poorly defined role of the actin nucleation regulators N-WASP and WASP in the DNA damage response. The manuscript uses a variety of approaches to carefully examine previously reported phenotypes of actin regulation in DNA repair, and the authors have done well to design controlled experiments. The manuscript thoroughly investigates N-WASP in U2OS cells, characterising cell cycle profiles, cell morphologies and studying DNA repair mechanisms and outcomes. A key component of this work was the use of DNA repair reporter assays which demonstrate the lack of a functional role for either N-WASP or WASP in homologous recombination (HR). This is in contradiction with previous reports, but I believe the authors have robustly designed and carried out their experiments and that the results are reliable, whereas the previous works have flaws that are unresolved in those publications. The manuscript also studies and discusses important points, such as the lack of WASP expression in most cell lines and the reliance on antibodies with low-specificity.

Overall, I believe this work to be of high quality and to address key points of interest within the current literature. Although limited in some aspects, the results presented are important and would make good contributions to the field. I would recommend publication after the following comments are addressed.

Major comments:

1. The authors have focused on the U2OS cell line for comparison with previously published work. I agree with this in principle, however for publication of the findings it would be very beneficial for an additional cell line to be tested that is distinct from U2OS. Otherwise, there is the potential for the results to be seen as U2OS specific, which I do not believe is the case, and which is not the message of the manuscript. Of course, only a few experiments that the authors deem key to their conclusions need to be confirmed in another cell line, not all experiments.

For the revised version we generated a new N-WASP KO U2OS cells with another gRNA and N-WASP KO MCF-10A and HEK293T cells. In all cell lines, loss of N-WASP did not affect the cell cycle and no potentially compensatory expression of WASP was observed (Fig. 1C, 1A). Performing a cell cycle resolved γ H2AX kinetic following irradiation, none of the N-WASP KO cell lines showed an obvious delay in DNA repair (Fig. 2) or a significant increase in micronuclei formation (Fig. 3). Both U2OS and HEK293T N-WASP KO cells showed no reduction in HDR or total gene editing of two different target genes in a IDAA-based assay with single-stranded repair templates (Fig. 7A). In the mClover fluorescent reporter assay, which uses a double stranded DNA repair template, N-WASP KO U2OS cells showed no alteration of HDR, while N-WASP KO HEK293T cells displayed a reduced HDR, suggesting some extent of cell type specificity (Fig. 7B). Notably, whereas U2OS cells exhibited similar HDR efficiencies across all three tested loci, the absolute HDR efficiency for LMNA in the mClover assay was only around 1%, which is markedly lower than the approximately 25% HDR efficiencies observed for PAX2 and TSPAN12 in the IDAA assay. We could not observe a significant increase of micronuclei formation in U2OS, HEK293T, and MCF-10A cells 24h after irradiation (Fig. 3). Treatment of U2OS and HEK293T N-WASP KO cells with PARPi did not cause a synthetic lethality effect, which is otherwise often observed in cells with a defect in HR genes (Fig. 7C). These results demonstrate that N-WASP is not having an important, general role in DNA repair, although cell type and target gene specific effects cannot be excluded.

With respect to WASP we showed that neither U2OS, nor HEK293T or MCF-10A cells express WASP (Fig. 1A), in line with earlier reports that WASP expression is restricted to the hematopoietic lineage. We therefore did not generate WASP KO and WASP overexpression models for other cells than U2OS.

2. Figure EV2C should have a WT comparison for the signal of the WASP antibody, as if the signal is 10-fold lower in the WASP KO cells then a argument can be made for specificity. Over-expression of the HA-tagged WASP functionally works as a control, and I believe the authors' interpretation of this, but fluorescence intensity is arbitrary and needs sufficient controls for appropriate comparisons. This should also be quantified.

We performed the experiment as suggested by the reviewer and quantified the stainings. Overexpression of HA-WASP in WASP KO U2OS cells resulted in a strong WASP signal in WB using the anti-WASP antibody (Fig. EV2B) and a strong immunofluorescence staining against the HA-tag in cultured cells (Fig. EV2C). Untransfected WASP KO U2OS cells showed no detectable WASP band using the anti-WASP antibody, similar to control transduced U2OS cells (Fig. EV2B). Immunofluorescence staining with the anti-WASP antibody, however, resulted in strong nuclear staining, which was not significantly altered in HA-WASP overexpressing cells. Quantification of the images confirmed the visual analysis (Fig. EV2D). Furthermore, HA-WASP showed mostly a cytoplasmic location as detected by anti HA antibodies, while the WASP antibody staining displayed a much higher nuclear fraction of staining (Fig. EV2E).

3. γ H2AX foci is only studied in the context of N-WASP KO. It would be very beneficial if WASP KO and possibly even CK-666 treatment was also studied in these experiments for comparison to N-WASP. The previous Shrank et al. manuscript observed delayed DSB repair following CK-666 treatment specifically in G2 cells and only with enzyme induced breaks via qPCR. It would be interesting to compare to this work.

We first investigated the effect of CK-666 treatment on the cell cycle of U2OS cells after irradiation. After 24h, we observed a significant increase in G1 phase, confirming the results observed in non-irradiated cells (Fig. 9A). However, the differences were less pronounced, suggesting that irradiation modified the effect of CK-666 on the cell cycle. We then determined the cell cycle resolved γ H2AX kinetic following irradiation. An increased frequency of γ H2AX foci was observed 1h after irradiation and a slightly increased mean fluorescence of nuclear γ H2AX in S phase after 6h (Fig. EV9 B, C). These relatively subtle alterations argue against a prominent role for Arp2/3 mediated actin polymerization in DNA repair.

Since we conducted the suggested experiments with CK-666 and convincingly showed that our U2OS do not express WASP and are therefore equivalent to WASP KO U2OS cells (Fig. EV2), we did not compare wild type U2OS and WASP KO U2OS in more assays than shown already in Fig. 5 and Fig. 6.

Minor comments:

1. Introduction states RAD51 enables then repair by Homologous Recombination, I believe this should be "then enables repair by" or something similar, I'm not sure if the current form is correct.

We corrected this sentence.

2. Figure 1 needs representative images.

We prepared a corresponding figure with example images (Fig. EV1) Access to all images analyzed is available through BioStudies.

3. Page 14 first line, should be tagging of N-WASP and WASP "with HA" not "by HA"

We corrected this sentence.

Referee #3:

Brakebusch and colleagues present evidence that in U2OS cells, N-WASP and WASP are not required to mediate HR-dependent repair of DSBs. Given dubious data presented in the highly cited Nature paper by Shrank et al suggesting that the haematopoietic-restricted

actin nucleating factor, WASP, plays an important role in regulating DSB repair in epithelial cells, it is important to publish this manuscript questioning these findings. However, since it is not clear to me that the N-WASP KO cell generated in this manuscript is truly null, it is difficult for the authors to make such a strong statement about the role of N-WASP in HR-directed repair. Despite this, with some revisions (detailed below), I believe that this manuscript would be suitable for publication in EMBO Reports.

Specific points:

1. Figure EV1A: the N-WASP CRISPR KO cell line does not look like a complete knockout as there is residual protein. Based on this, it is difficult to conclude whether the lack of a DSB repair, cell migration or replication defect in these cells is due to the presence of residual protein being expressed. Can the residual N-WASP be reduced with siRNA? Does depletion of N-WASP in U2OS cells affect DSB repair or cell migration?

To improve the N-WASP KO efficiency we tested several additional sgRNAs and identified one with a better KO efficiency than the sgRNA of our original submission (Fig. 1A). Indeed, the U2OS cells with an improved KO efficiency showed a slight defect in cell migration, as the reviewer expected (Fig. 1B).

Similar to the earlier results, we observed with the new N-WASP KO cells an unaltered cell cycle (Fig. 1C) and unaltered nuclear γ H2AX foci number and mean fluorescence per pixel in an irradiation induced cell cycle resolved kinetic (Fig. 2A, B) (Mean fluorescence is a very robust parameter, since it is not dependent on the identification of foci by an image analysis program, which could be affected by altered size or intensity of foci).

Moreover, HDR efficiency of N-WASP KO U2OS was not decreased in an IDAA-based assay against 2 targets (single stranded repair template; Fig. 7A) and in a fluorescence reporter based assay (mClover; double-strand reporter template; Fig. 7B). In the PAX2 IDAA-assay with U2OS, HDR was increased in N-WASP KO, which might be related to the increased KO efficiency.

2. Figure 1: 10Gy is a very high dose of ionising radiation. Since it is well established that defects in MRN-dependent processes are better observed following exposure to low doses of IR, it may be that a defect in the resolution of H2AX foci in the N-WASP 'KO' cells is only detectable following exposure to 1-2Gy of IR. However, since IR-induced DSBs are predominantly repaired by NHEJ, it might be better to examine the presence of a DSB repair defect in the N-WASP 'KO' cells using camptothecin, where the DSBs are exclusively repaired by HR.

For the revised version, we repeated all irradiation experiments with 2 Gy instead of 10 Gy. As with 10Gy, we did not see with 2 Gy a significant alteration of the frequency of γ H2AX foci or of the mean nuclear γ H2AX intensity (Fig. 2). In the revised manuscript, we show now only the 2 Gy results.

We used camptothecin to induce DNA breaks for the co-localization experiment. Here we found partial colocalization of γ H2AX with BRCA1, but not with HA-WASP or HA-N-WASP (Fig. 4C). Moreover, Triton-extraction efficiently removed nuclear HA-NWASP, but not camptothecin induced γ H2AX foci (Fig. 4D). These results do not support a role for N-WASP at DSB repaired by HR.

3. Figure 2: Many proteins involved in the repair of DSBs are not observed as relocating to foci using conventional immunofluorescence protocols. This does not mean they are not present at DSBs. It would be more prudent to assess the localisation of N-WASP to DSBs using PLA with γ -H2AX or using the Fok1-induced DSB system generated by the Greenberg lab. This approach has previously been used to show the presence of DIAPH1, another actin nucleating factor, at DSBs (Woodward et al. 2025. Nature Commun. 16:4491).

We optimized our immunofluorescence staining protocol for γ H2AX and HA allowing us to clearly detect nuclear foci of HA-N-WASP. Pearson's correlation coefficient, however, did not detect any colocalization of HA-N-WASP or HA-WASP with camptothecin induced γ H2AX foci, while γ H2AX strongly colocalized with BRCA1 (Fig. 4C). High specificity of the HA staining was proven by staining of WASP KO and HA-WASP overexpressing U2OS cells, which showed virtually no background in WASP KO cells (Fig. EV2C). High specificity of the γ H2AX staining was demonstrated by the high increase of γ H2AX staining following treatment with irradiation or camptothecin (fx. Fig 4D). These data suggest that WASP or N-WASP are not co-localizing with DNA breaks.

Using the optimized antibody binding conditions from the immunofluorescence staining we then conducted a PLA assay using the Duolink kit. However, we unfortunately obtained only background stain. While it is possible to titrate now different antibody concentrations, we decided not to pursue this line, since the data described above argue at least against a strong binding of N-WASP at DNA double-strand breaks. Furthermore, immunofluorescence was the methodology used earlier by Schrank et al to demonstrate localization of WASP at DSB. We mention in the discussion of the revised manuscript that the presence of a very low amount of N-WASP at the DSBs cannot be excluded and describe the data of Woodward et al.

4. Figure EV2: The inability to detect WASP in U2OS cells by Western blotting is consistent with WASP expression being predominantly limited to haematopoietic cells and puts into question the data demonstrating a role for WASP in promoting DSB repair published in the manuscript by Schrank et al. 2018. This is an important observation that should be published.

In line with WASP expression restricted to the hematopoietic system, we show in the revised manuscript that we could not detect WASP protein by Western blot in MCF10A and HEK293T cells (Fig. 1A).

5. Figure 3: A positive control should be carried out for the HR and NHEJ assays i.e. either knocking down or inhibiting RAD51 and DNA ligase IV.

Treating U2OS cells with the RAD51 inhibitor B02 significantly reduced HDR efficiency in the IDAA-based DNA repair assay (Fig. for reviewers 1). This repair assay involves a single stranded DNA repair template for HDR, which was reported to be mediated by a RAD52 dependent mechanism, which might contribute to the incomplete inhibition of HDR by B02.

Fig. for reviewers 1: RAD51 inhibitor B02 decreases HDR efficiency in U2OS cells as measured by targeting PAX2 and TSPAN12.

Earlier, we confirmed the validity of our IDAA-based assay system by comparing it with Traffic Light reporter assays and NGS based analyses (Bischoff et al; doi.org/10.1016/j.ggedit.2022.100023). Finally, we used now also the mClover fluorescence reporter system to confirm that N-WASP KO U2OS cells show unaltered HDR (Fig. 4B).

To control for the cell cycle artifacts induced by CK-666 treatment, using immunofluorescence and co-staining with a marker of the cell cycle e.g. CENPF or Cyclin A/B will allow quantification of HR in S/G2 cells only.

To address this question we performed now a cell cycle resolved irradiation induced γ H2AX kinetic of U2OS cells in the presence and absence of CK-666, revealing subtle CK-666 dependent alterations, but not an obvious cell cycle phase specific role of Arp2/3 (Fig. 9). We could not directly use staining of cell cycle markers in DNA repair assay, since the fixation for the staining will crosslink the DNA, and therefore prevents its use for later PCR in the IDAA-based assay. Secondly, in the IDAA and the mClover assay, cells are investigated at a time point long after the DNA repair has taken place. Therefore, staining at the collection time will not reflect exactly the cell cycle phase the cells were at the time of DSB induction and repair.

Furthermore, knocking down a component of the ARP2/3 complex with siRNA would be a useful control. This has been recently shown by Woodward et al. 2025. (Nature Commun. 16:4491).

We knocked down ARPC4 in U2OS cells and demonstrated a significant alteration of cell cycle (Fig. 8B) and a reduction of HDR using mClover assay system (Fig. 10C). These results are similar to the CK-666 results.

6. Figure 4: It was previously demonstrated by Woodward et al. that loss of DIAPH1 gives rise to defective HR, which was more prevalent when assaying it using the CRISPR-based mClover-LMNA HR assay developed by Graham Delaire than using the DR-GFP reporter cell

line. This indicates that the type of reporter assay used to measure involvement of the role of the actin nucleation pathway in DSB repair is important i.e. the type of DSB generated may dictate how much inhibiting the actin nucleation pathway affects HR-dependent repair.

Following the excellent suggestion by the reviewer we used now also the mClover assay. As with the IDAA-based assay we observed unaltered HDR efficiency in N-WASP KO U2OS cells (Fig. 7B). In the newly established N-WASP KO HEK293T cells, however, the IDAA-based assay showed unaltered HDR efficiency when targeting PAX2 or TSPAN12 (Fig. 7A), while HDR efficiency in the mClover assay, which targets LMNA, was reduced (Fig. 7B). In addition to the targeting of different genomic locations, also mechanistical differences in the respective DNA repair mechanisms might be involved, since the IDAA-based assay is using single stranded DNA repair templates, while the mClover assay is relying on a double-stranded DNA repair plasmid. Notably, whereas U2OS cells exhibited similar HDR efficiencies across all three tested loci, the absolute HDR efficiency for LMNA in the mClover assay was only around 1%, which is markedly lower than the approximately 25% HDR efficiencies observed for PAX2 and TSPAN12 in the IDAA assay.

7. The recent publication by Woodward et al. demonstrating a role for DIAPH1 and gamma-actin regulating DSB repair should be cited. This manuscript shows that manipulating the ability of cells to nucleate actin through the use of naturally occurring human mutations rather than non-specific chemical inhibitors can affect HR. Moreover, this manuscript is consistent with the authors indicating that other actin nucleating factors, rather than WASP/N-WASP, are important for DSB repair.

In the revised manuscript, we cite this important paper and shortly discuss its findings.

For all reviewers:

Raw data are accessible at BioStudies (the BioImages accession number S-BIAD2513) at: <https://www.ebi.ac.uk/biostudies/bioimages/studies/S-BIAD2513?key=e400e1e1-e4ff-48af-8b76-e03df52b0d72>

Dear Prof. Brakebusch,

Thank you for the submission of your revised manuscript. We have now received the enclosed reports from the referees as well as referee cross-comments that I attach to this email.

As you will see, referee 1 is not satisfied with the revised ms, but both referees 2 and 3 agree that your data are sufficiently strong and important and should be published. We can therefore proceed with acceptance after the last minor revisions will have been performed. Please submit a point-by-point response with your final ms.

- The Figure legends need to be placed at the end of the ms file.
- The Data Availability Section should be placed before the Acknowledgement section.
- Please remove the Author Contributions from the ms file. All contributions need to be entered during online ms submission.
- In the author checklist, 2 responses are missing, please chose a response from the drop down menus and submit a completed author checklist. All questions on statistics need to be answered.
- The FUNDING INFO, the Grant R302-A17455 Danish Cancer Society Research Center (DCRC), is listed in our online submission system but not included in the manuscript, please also add it to the funding info in the ms file.
- Please upload all figures as separate figure files.
- FIGURE CALLOUTS are missing for Figure 4A-B and Figure EV1, please add.
- The Reagents & Tools TABLE needs to be removed from the ms file and uploaded as a separate Reagents table file.
- Materials and Methods section to be relabelled as 'Methods'
- Figure 6: label A missing in the figure caption, please add.
- Figure 8: label C missing in the figure caption, please add.

* Figure Legends - Comments *

- Please note that the legend for figure 6A, 8C is missing in the manuscript. This needs to be rectified.
- Please note that the exact p values are not provided in the legends of figures 1A, B; 2A, B; 4C, D; EV2 D, E; 5A-C; 6A, 7A-C; EV3 A, B; 8 A, B, C; 9A-C; 10A-C. Please provide exact values as reasonable.

EMBO press papers are accompanied online by A) a short (1-2 sentences) summary of the findings and their significance, B) 2-3 bullet points highlighting key results and C) a synopsis image that is exactly 550 pixels wide and 200-600 pixels high (the height is variable). The synopsis image should provide a sketch of the major findings, like a graphical abstract. Please note that text needs to be readable at the final size. Please send us this information along with the final manuscript.

Referee #1:

I appreciate the effort the authors have made to address the concerns raised in the first round of revision. The revised manuscript includes additional cell lines with improved KO efficiency, new attempts at IF staining with Triton extraction, a second

HR reporter assay (mClover), ARPC4 knockdown, lower IR doses. Overall, the study is improved compared to the original version. However, my overall assessment remains unchanged. This study does not add much to our current knowledge, thus I do not recommend publication in EMBO Reports.

The main issue remains that it is much more difficult to prove the absence of a phenotype than its presence. The experiments proposed also still do not match the breadth of approaches used in previous studies that reported recruitment and functional involvement of actin polymerization through WASP and Arp2/3 (e.g. WASP enrichment in damaged chromatin by pulldown or ChIP at defined DSB sites, well-established HDR reporter assays, quantification of repair foci in cell cycle arrested cells, analysis of the dynamic response of HR repair foci, etc). This study also uses distinct approaches from what published (e.g., CRISPR-mediated repair assays rather than GFP reporters, different antibody concentration, repair kinetics in non-arrested cells etc). If a study aims to disprove previous evidence, at the very least the same reagents and conditions should be applied and if data are not reproduced it is critical to understand the reason why differences are observed in the first place before moving to other approaches.

Main points:

- The absence of major effect in γ H2AX foci kinetics after N-WASP removal could derive from other compensatory repair pathways taking over.

- The absence of colocalization by IF does not definitively exclude recruitment. As also mentioned by another reviewer, many repair proteins do not form clear repair foci, and amplification techniques might need to be used to detect signals. This includes laser treatments, ChIP assays, etc. Antibody variability (e.g., lot differences, maintenance) or staining variables are common issues. We also notice that previous studies used a much higher concentrations of the anti-WASP antibody mentioned here (e.g., Shwartz paper). Furthermore, as noticed by the authors this antibody clearly detects a lot of aspecific signals at the conditions used for Wb (EV2B), and highly stringent washes, different conditions of fixation, or higher amount of antibody might need to be applied to increase the signal to noise ration and minimize aspecific signals. Another reviewer urged to compare WT WASP with WASP KO to validate the staining (not just HA-WASP over-expressing cells), and this is still not done in the revised manuscript. This is a very basic and necessary control for any IF. Similarly, PLA experiments were not optimized and not shown with proper controls, thus they cannot be evaluated in revision and they remain inconclusive.

- Regarding HDR assays, the inclusion of both IDAA- and mClover-based systems is appreciated. However, differences in repair template type (ssDNA vs dsDNA), genomic loci, and HDR efficiencies relative to the DR-GFP complicate direct comparison with prior studies. WASP could have been expressed at higher level in cells used in previous studies, hence have a major role in repair. The reduced HDR observed in HEK293T in the mClover assay (Fig. 7B) further suggests possible context dependency of the HDR defects rather than universal dispensability. A similar comment refers to HDR defects observed in U2OS cells after CK66 treatment.

- Regarding CK666, a 24h pretreatment is very uncommon and not what done in previous studies (e.g., Shrank et al). A few minutes up to to 1h is typically used and then cells are damaged, thus more likely become arrested. I find the experiments done with long CK66 treatments and the conclusions driven deceiving.

- Effects of ARPC4 on the cell cycle seem minimal.

- This study also shows cell cycle effects and focus kinetic quantifications also after 1h treatments, which results in minimal effects on the cell cycle. However, it is not clear whether these treatments are working in terms of disrupting Arp2/3 function (no controls provided) and particularly nuclear Arp2/3 or nuclear actin polymerization. Without controls it is difficult to explain discrepancies with previous studies. Of note, previous studies showed HR defects after 6 and 12 hr treatment with CK666 (Shrank et al) in G2 cells. The kinetics provided here show virtually no cell cycle effects in G2 at these timepoints, thus again the cell cycle effects do not seem very relevant to the interpretation of previous data.

- The effect on a 1h CK66 treatment on transfection efficiency is an interesting result. Transfection efficiency needs to be tested for every protein and RNAi under investigation in different treatment conditions, as variabilities occur for all sort of reasons.

In summary, the revised manuscript does not disprove that Arp2/3 and WASP/N-WASP contributed to HR repair in human cells - though this effect might be cell line and context specific. There is no effort to directly replicate the experimental approaches previously published to identify potential discrepancies, and many controls are missing. Yet the authors continuously refer to previous studies in an attempt to disprove these findings. Given these major concerns, I do not recommend publication in EMBO Reports.

Other issues

It is stated that "WASP antibody was able to detect genuine WASP, as suggested by its overlapping with HA signals", but I could not find this result

Referee #2:

The revised manuscript stands greatly improved. Although the outcomes and conclusions are consistent with the previous version, the supporting data is significantly more broad and deep. I believe the data presented by the authors definitively show many important phenotypes, or lack thereof, in the regulation of DNA repair. This includes the lack of impact of N-WASP/WASP on cell cycle, homologous recombination and genome stability. This is demonstrated across multiple cell lines and methods of depletion/inhibition and in a wide variety of assays to study each phenotype.

To support the many negative phenotypes, the authors have validated gene expression, knockout efficiency and antibody specificity as well as testing multiple cell lines.

Many results here are contradictory to those published in a previous manuscript by Shrank et al.; however, the findings in this revised manuscript are far more robust than those in the previous. The findings presented here are of great importance for the DNA repair field, particularly as the roles of N-WASP/WASP are not well expanded upon from the Shrank et al. manuscript in 2018. This work could therefore form an important basis for this area of DNA repair going forward.

I recommend this manuscript be accepted for publication in its current state.

Referee #3:

The authors have adequately addressed my comments. I would recommend this manuscript for publishing.

Note: one of the histograms in Figure 7B lacks labels.

Referee #1

I appreciate the effort the authors have made to address the concerns raised in the first round of revision. The revised manuscript includes additional cell lines with improved KO efficiency, new attempts at IF staining with Triton extraction, a second HR reporter assay (mClover), ARPC4 knockdown, lower IR doses. Overall, the study is improved compared to the original version. However, my overall assessment remains unchanged. This study does not add much to our current knowledge, thus I do not recommend publication in EMBO Reports.

The main issue remains that it is much more difficult to prove the absence of a phenotype than its presence. The experiments proposed also still do not match the breadth of approaches used in previous studies that reported recruitment and functional involvement of actin polymerization through WASP and Arp2/3 (e.g. WASP enrichment in damaged chromatin by pulldown or ChIP at defined DSB sites, well-established HDR reporter assays, quantification of repair foci in cell cycle arrested cells, analysis of the dynamic response of HR repair foci, etc). This study also uses distinct approaches from what published (e.g., CRISPR-mediated repair assays rather than GFP reporters, different antibody concentration, repair kinetics in non-arrested cells etc). If a study aims to disprove previous evidence, at the very least the same reagents and conditions should be applied and if data are not reproduced it is critical to understand the reason why differences are observed in the first place before moving to other approaches.

Whereas KO cell lines can adapt over time, Figure 1B demonstrates significantly reduced wound-closure rate, indicating the cells present with an expected phenotype.

Main points:

- The absence of major effect in γ H2AX foci kinetics after N-WASP removal could derive from other compensatory repair pathways taking over.

γ H2AX levels were also unchanged with short term treatment with the ARP2/3 inhibitor CK666. The original manuscript did not assay γ H2AX foci kinetics which is actually a major missing component of a study of novel DNA repair mechanisms.

- The absence of colocalization by IF does not definitively exclude recruitment. As also mentioned by another reviewer, many repair proteins do not form clear repair foci, and amplification techniques might need to be used to detect signals. This includes laser treatments, ChIP assays, etc. Antibody variability (e.g., lot differences, maintenance) or staining variables are common issues. We also notice that previous studies used a much higher concentrations of the anti-WASP antibody mentioned here (e.g., Shwartz paper). Furthermore, as noticed by the authors this antibody clearly detects a lot of aspecific signals at the conditions used for Wb (EV2B), and highly stringent washes, different conditions of fixation, or higher amount of antibody might need to be applied to increase the signal to noise ration and minimize aspecific signals. Another reviewer urged to compare WT WASP with WASP KO to validate the staining (not just HA-WASP over-expressing cells), and this is still not done in the revised manuscript. This is a very basic and necessary control for any IF. Similarly, PLA experiments were not optimized and not shown with proper controls, thus they cannot be evaluated in revision and they remain inconclusive.

The authors here attempted to validate antibody specificity and found the WASP antibody to be completely non-specific in IF as WASP KO cells had the same signal intensity as WASP overexpression cells. The fact that this is the same antibody used in the previous

publication is a point of concern, the fact that the previous publication also used a higher concentration of the antibody only makes this point worse as this would increase non-specific binding. The authors used the same fixation as the original manuscript and in my opinion, the non-specific signal demonstrated here is not an issue of signal:noise as there is really no on-target signal detected, I do not believe optimisation of staining or washing conditions can improve this and no specific conditions have been mentioned in the previous publication.

As for the WT control for the immunofluorescence, I believe my original point has been addressed appropriately. The authors pointed out they conducted this experiment in western blot form and quantified their immunofluorescence as requested. In their quantification, they show that the WASP antibody signal is mostly nuclear and this is unchanged upon overexpression, therefore demonstrating that it is nothing to do with signal:noise or absolute signal levels as I had theorised.

The experiments conducted by the authors here clearly show that the WASP antibody is non-specific and unsuitable for these experiments. In addition, they show their overexpressed construct produces specific signal, but no localisation to sites of DNA damage.

-Regarding HDR assays, the inclusion of both IDAA- and mClover-based systems is appreciated. However, differences in repair template type (ssDNA vs dsDNA), genomic loci, and HDR efficiencies relative to the DR-GFP complicate direct comparison with prior studies. WASP could have been expressed at higher level in cells used in previous studies, hence have a major role in repair. The reduced HDR observed in HEK293T in the mClover assay (Fig. 7B) further suggests possible context dependency of the HDR defects rather than universal dispensability. A similar comment refers to HDR defects observed in U2OS cells after CK66 treatment.

This isn't an unreasonable comment, although the presence of a phenotype in one of the six such conditions tested in Figure 6 is not very convincing of context dependent phenotypes.

- Regarding CK666, a 24h pretreatment is very uncommon and not what done in previous studies (e.g., Shrank et al). A few minutes up to to 1h is typically used and then cells are damaged, thus more likely become arrested. I find the experiments done with long CK66 treatments and the conclusions driven deceiving.

The authors do not state a 24-hour pre-treatment. In experiments where the cells were treated with DNA damaging agents a 1 hour pre-treatment was used (Figure 9). The authors used 24 hour treatment to demonstrate that CK-666 effects the cell cycle distribution of cells in experiments that require such a long treatment, such experiments include the HDR reporter assays discussed extensively in the manuscript as these experiments take long periods for repair to complete. The previous Shrank et al publication appeared to use a 48-hour treatment.

- Effects of ARPC4 on the cell cycle seem minimal.

This is true, there are multiple possible explanations though I don't think this warrants much further investigation.

- This study also shows cell cycle effects and focus kinetic quantifications also after 1h

treatments, which results in minimal effects on the cell cycle. However, it is not clear whether these treatments are working in terms of disrupting Arp2/3 function (no controls provided) and particularly nuclear Arp2/3 or nuclear actin polymerization. Without controls it is difficult to explain discrepancies with previous studies. Of note, previous studies showed HR defects after 6 and 12 hr treatment with CK666 (Shrank et al) in G2 cells. The kinetics provided here show virtually no cell cycle effects in G2 at these timepoints, thus again the cell cycle effects do not seem very relevant to the interpretation of previous data.

The authors do not state that cell-cycle defects is the explanation of the HDR defects observed with CK-666 treatments. Instead it is suggested as a potential factor alongside others, particularly transfection efficiency. The authors could more clearly phrase this in the discussion.

- The effect on a 1h CK66 treatment on transfection efficiency is an interesting result. Transfection efficiency needs to be tested for every protein and RNAi under investigation in different treatment conditions, as variabilities occur for all sort of reasons.

The authors are assaying transfection efficiency as the HDR reporter assays use transfection of the reporter in one way or another, there is a chance that a GFP plasmid will behave different to the mClover HDR assay plasmid, this is quite unlikely.

In summary, the revised manuscript does not disprove that Arp2/3 and WASP/N-WASP contributed to HR repair in human cells - though this effect might be cell line and context specific. There is no effort to directly replicate the experimental approaches previously published to identify potential discrepancies, and many controls are missing. Yet the authors continuously refer to previous studies in an attempt to disprove these findings. Given these major concerns, I do not recommend publication in EMBO Reports.

The manuscript here is in many ways more robust than the previous Shrank et al manuscript. The authors have made clear efforts to get to the bottom of the phenotypes, and lack thereof, observed. From my perspective, the manuscript appears more as set of work that was initially conducted to study these phenotypes, which turned into the in-depth analysis it is due to the challenge the authors found in trying to replicate the results of others.

In my opinion, regardless of if one or another publication is "correct", the publication of results conflicting with previous data is imperative to the integrity of scientific literature. Discounting a manuscript primarily because it is in conflict with a previous publication is not in the interest of the scientific community.

Other issues

It is stated that "WASP antibody was able to detect genuine WASP, as suggested by its overlapping with HA signals", but I could not find this result

Referee #1

I appreciate the effort the authors have made to address the concerns raised in the first round of revision. The revised manuscript includes additional cell lines with improved KO efficiency, new attempts at IF staining with Triton extraction, a second HR reporter assay (mClover), ARPC4 knockdown, lower IR doses. Overall, the study is improved compared to the original version. However, my overall assessment remains unchanged. This study does not add much to our current knowledge, thus I do not recommend publication in EMBO Reports.

The main issue remains that it is much more difficult to prove the absence of a phenotype than its presence. The experiments proposed also still do not match the breadth of approaches used in previous studies that reported recruitment and functional involvement of actin polymerization through WASP and Arp2/3 (e.g. WASP enrichment in damaged chromatin by pulldown or CHIP at defined DSB sites, well-established HDR reporter assays, quantification of repair foci in cell cycle arrested cells, analysis of the dynamic response of HR repair foci, etc). This study also uses distinct approaches from what published (e.g., CRISPR-mediated repair assays rather than GFP reporters, different antibody concentration, repair kinetics in non-arrested cells etc). If a study aims to disprove previous evidence, at the very least the same reagents and conditions should be applied and if data are not reproduced it is critical to understand the reason why differences are observed in the first place before moving to other approaches.

Comment: I agree with the reviewer that it is much more difficult to prove the absence of a phenotype than its presence. However, if the robustness of a study is judged by whether it can be recapitulated by other labs using similar or alternative/orthogonal approaches, then it should not matter whether the authors have used identical reagents to those published by Shrank et al or not, the phenotype should still be present. The primary problem with the Shrank et al paper is that WASP is only expressed in haematopoietic cells and not in U2OS cells as was published by Shrank et al. Therefore the ability of the authors to recapitulate this data using the same reagents will not be possible.

Main points:

- The absence of major effect in γ H2AX foci kinetics after N-WASP removal could derive from other compensatory repair pathways taking over.

Comment: This comment is unfair. It is more likely that the absence of a major effect on H2AX foci kinetics when N-WASP is lost is because N-WASP doesn't play a role in DSB repair rather than being due to the existence of some 'hypothetical' compensatory pathway taking over, the latter of which will be difficult to prove.

- The absence of colocalization by IF does not definitively exclude recruitment. As also mentioned by another reviewer, many repair proteins do not form clear repair foci, and amplification techniques might need to be used to detect signals. This includes laser treatments, ChIP assays, etc. Antibody variability (e.g., lot differences, maintenance) or staining variables are common issues. We also notice that previous studies used a much higher concentrations of the anti-WASP antibody mentioned here (e.g., Shwartz paper). Furthermore, as noticed by the authors this antibody clearly detects a lot of aspecific signals at the conditions used for Wb (EV2B), and highly stringent washes, different conditions of fixation, or higher amount of antibody might need to be applied to increase the signal to noise ration and minimize aspecific signals. Another reviewer urged to compare WT WASP with WASP KO to validate the staining (not just HA-WASP over-expressing cells), and this is still not done in the revised manuscript. This is a very basic and necessary control for any IF. Similarly, PLA experiments were not optimized and not shown with proper controls, thus they cannot be evaluated in revision and they remain inconclusive.

Comment: Again I agree with the reviewer that an absence of colocalization by IF does not definitely exclude recruitment and did suggest the authors carry out PLA as an alternative method to assess whether WASP/N-WASP are recruited to sites of DNA damage. Whilst this suggestion was not carried out by the authors, I do concede that PLA is an expensive and time consuming technique where the benefits of proving a negative might not outweigh the considerable time it might take to optimise this protocol.

-Regarding HDR assays, the inclusion of both IDAA- and mClover-based systems is appreciated. However, differences in repair template type (ssDNA vs dsDNA), genomic loci, and HDR efficiencies relative to the DR-GFP complicate direct comparison with prior studies. WASP could have been expressed at higher level in cells used in previous studies, hence have a major role in repair. The reduced HDR observed in HEK293T in the mClover assay (Fig. 7B) further suggests possible context dependency of the HDR defects rather than universal dispensability. A similar comment refers to HDR defects observed in U2OS cells after CK66 treatment.

Comment: It has been previously shown that loss of the actin-dependent repair pathway does give rise to some variability when using the DR-GFP vs mClover-CRISPR HR assays, with the latter showing a more robust and reproducible reduction in HR when components of the actin polymerisation pathway are depleted (Woodward et al, 2025, Nature Commun). Therefore, as reviewer 1 highlights, the context of the DSB does matter and that the DR-GFP assay is not a good assay for measuring mild to moderate HR defects. However, I am not really sure what the reviewer is criticizing here.

- Regarding CK666, a 24h pretreatment is very uncommon and not what done in previous studies (e.g., Shrank et al). A few minutes up to to 1h is typically used and then cells are damaged, thus more likely become arrested. I find the experiments done with long CK66 treatments and the conclusions driven deceiving.

Comment: The CK666 inhibitor is very toxic to cells, so I agree that a 24h incubation time seems unnecessarily long. I guess if the authors did not see an effect of CK666 on DNA repair after 1h, then longer timepoints would be the obviously next step. However, I am not sure what conclusions made from data generated using longer incubations with CK666 would be deceiving.

- Effects of ARPC4 on the cell cycle seem minimal.

Comment: No comment necessary.

- This study also shows cell cycle effects and focus kinetic quantifications also after 1h treatments, which results in minimal effects on the cell cycle. However, it is not clear whether these treatments are working in terms of disrupting Arp2/3 function (no controls provided) and particularly nuclear Arp2/3 or nuclear actin polymerization. Without controls it is difficult to explain discrepancies with previous studies. Of note, previous studies showed HR defects after 6 and 12 hr treatment with CK666 (Shrank et al) in G2 cells. The kinetics provided here show virtually no cell cycle effects in G2 at these timepoints, thus again the cell cycle effects do not seem very relevant to the interpretation of previous data.

Comment: I am not sure that the Shrank paper showed any evidence that the treatments of CK666 they used disrupted Arp2/3 function and/or nuclear actin polymerisation. CK666 is a well characterised inhibitor of the ARP2/3 complex used by many groups, so while it would be nice to show that it is inhibiting actin polymerisation at the concentrations stated, I don't think this is essential.

- The effect on a 1h CK66 treatment on transfection efficiency is an interesting result. Transfection efficiency needs to be tested for every protein and RNAi under investigation in different treatment conditions, as variabilities occur for all sort of reasons.

Comment: Since CK666 was only used for two figures (figures 8 and 10), which focus on assessing whether inhibition of the ARP2/3 complex affects DSB repair where transfection efficiency has to be controlled for using plasmid encoding a fluorescent protein, I don't think this comment is valid. Protein levels in knockout cells and cells treated with siRNA have been verified.

In summary, the revised manuscript does not disprove that Arp2/3 and WASP/N-WASP contributed to HR repair in human cells - though this effect might be cell line and context specific. There is no effort to directly replicate the experimental approaches previously published to identify potential discrepancies,

and many controls are missing. Yet the authors continuously refer to previous studies in an attempt to disprove these findings. Given these major concerns, I do not recommend publication in EMBO Reports.

Comment: As stated above, if published results can't be recapitulated using alternative or orthogonal approaches, then this raises concerns about the validity of these results. I don't think that a lack of the authors trying to directly replicate the experimental approaches previously published using identical reagents would enhance this manuscript and would waste a lot of time, money and effort. Since that WASP is only expressed in haematopoietic cells and not in U2OS cells as previously published, how can it regulate DSB repair in epithelial cells? Given that the Shrank paper is being highly cited within the DSB repair field, it is important to publish findings that question the validity of these published results.

Other issues

It is stated that "WASP antibody was able to detect genuine WASP, as suggested by its overlapping with HA signals", but I could not find this result

All minor editorial requests have been addressed by the authors.

Prof. Cord Brakebusch
University of Copenhagen
BRIC
BRIC
Ole Maaløes Vej 5
Copenhagen 2200
Denmark

Dear Prof. Brakebusch,

I am very pleased to accept your manuscript for publication in the next available issue of EMBO reports. Thank you for your contribution to our journal.

You may qualify for financial assistance for your publication charges - either via a Springer Nature fully open access agreement or an EMBO initiative. Check your eligibility: <https://link.springer.com/journal/44319/how-to-publish-with-us>

>>> Please note that it is EMBO Reports policy for the transcript of the editorial process (containing referee reports and your response letter) to be published as an online supplement to each paper. If you do NOT want this, you will need to inform the Editorial Office via email immediately. More information is available here: <https://link.springer.com/partners/embo-press/editorial-policies#Peer%20review>